# Graph Random Features for Scalable Gaussian Processes

**Matthew Zhang**[1] **Jihao Andreas Lin**[1,2] **Krzysztof Choromanski**[3]
**Adrian Weller**[1,4] **Richard E. Turner**[1,4] **Isaac Reid**[1,3]

[1]University of Cambridge [2]Max Planck Institute for Intelligent Systems
[3]Google DeepMind [4]Alan Turing Institute
`matthew.zhang473@gmail.com ir337@cam.ac.uk`

## Abstract

We study the application of *graph random features* (GRFs) – a recently-introduced stochastic estimator of graph node kernels – to scalable Gaussian processes on discrete input spaces. We prove that (under mild assumptions) Bayesian inference with GRFs enjoys $\mathcal{O}(N^{3/2})$ time complexity with respect to the number of nodes $N$, with probabilistic accuracy guarantees. In contrast, exact kernels generally incur $\mathcal{O}(N^3)$. Wall-clock speedups and memory savings unlock Bayesian optimisation with over 1M graph nodes on a single computer chip, whilst preserving competitive performance.

## 1 Introduction and related work

Gaussian processes (GPs) provide a powerful framework for learning unknown functions in the presence of uncertainty (Rasmussen, 2003). In certain applications, kernels based on Euclidean distance may be unsuitable: for example, when modelling traffic congestion, since pairs of locations that are spatially close may not be connected by roads. In this case, kernels defined on the nodes of a graph $\mathcal{G}$ may be more appropriate (Borovitskiy et al., 2021; Smola and Kondor, 2003). One can then perform inference and make principled predictions, including during Bayesian optimisation, using GPs on graphs.[1]

**Scalability of GPs on graphs**. Like their Euclidean cousins, exact GPs on graphs incur $\mathcal{O}(N^3)$ time complexity with respect to the number of nodes $N$. This makes them impractical when working with very large graphs. To mitigate this, practitioners use techniques such as 'graph Fourier features', which approximate the kernel matrix with a truncated eigenvalue expansion, or specific sparse kernel families (Borovitskiy et al., 2021). The former loses high-frequency kernel information and the latter limits flexibility. Alternatively, one can use kernels for small, local subgraphs, at the cost of no longer performing inference on the whole of the graph $\mathcal{G}$ (Wan et al., 2023).

**Graph random features**. In this paper, we propose to instead use the recently-introduced class of *graph random features* (GRFs) – sparse, unbiased estimates of graph node kernels computed using random walks (Choromanski, 2023; Reid et al., 2023). GRFs are Monte Carlo estimators of power series of weighted adjacency matrices, analogous to Von Neumann's celebrated Russian Roulette estimator (Carter and Cashwell, 1975; Hendricks and Booth, 2006). GRFs enjoy strong concentration properties (Reid et al., 2024b). They are able to estimate a flexible class of graph node kernels – including the popular diffusion and Matérn kernels – by varying the so-called 'modulation function'.

**GRFs for scalable GPs**. Reid et al. (2024a) previously suggested using GRFs for GPs, as part of a broader study of variance reduction techniques. However, their experiments focused exclusively on the diffusion kernel with small graphs, failing to exploit the estimator's sparsity to accelerate inference. Moreover, they limited their (chiefly theoretical) study to computing the posterior, omitting exploration of applications such as Bayesian optimisation.

---

[1]We consider GPs defined on the nodes of a *fixed* graph. The input space is finite and we perform inference for a finite set of random variables, one per node. The relationships between these variables are determined by the structure of $\mathcal{G}$ via a graph node kernel. Whilst some might prefer to call this a 'Gaussian random field' or simply a 'multivariate Gaussian', in this paper we use 'GP on a graph' for consistency with recent literature (Borovitskiy et al., 2021; Reid et al., 2024a; Wan et al., 2023).

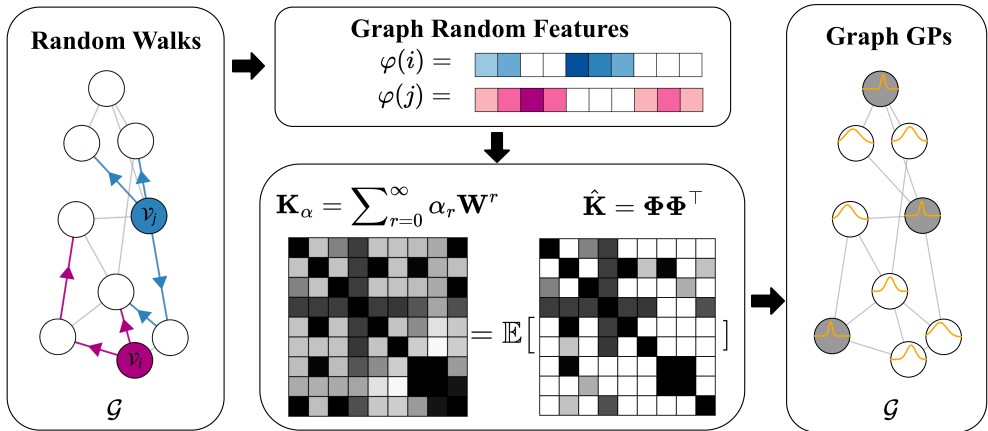

Figure 1: **GRFs for scalable GPs on graphs.** The GRF algorithm constructs random feature $\varphi(i)$ for node $i \in \{1, ..., N\}$ using random walks. $\hat{\mathbf{K}} := \left[\varphi(i)^\top \varphi(j)\right]_{i,j=1}^N$ is a sparse approximation of the kernel matrix $\mathbf{K}_\alpha$, enabling efficient posterior inference in $\mathcal{O}(N^{3/2})$.

**Key contributions.** We investigate graph random features (GRFs) for Gaussian processes (GPs), unlocking scalable Bayesian inference on graphs with $> 1\text{M}$ nodes. See Figure 1.

1. We use GRFs to construct sparse estimates of learnable graph node kernels, and use these as covariance functions for GPs.

2. We prove that Bayesian inference with GRFs enjoys $\mathcal{O}(N^{3/2})$ time complexity with probabilistic guarantees on approximation quality, compared to $\mathcal{O}(N^3)$ for exact alternatives. In experiments, this translates to $50\times$ wall-clock speedups on graphs with fewer than 10K nodes. Remarkably, the flexibility of GRFs sometimes enables them to outperform dense alternatives on test negative log probability density and root mean squared error.

3. We showcase our new techniques by performing Bayesian optimisation on massive graphs, implementing Thompson sampling with $> 1\text{M}$ nodes on a single computer chip.[2]

## 2 PRELIMINARIES

Consider an undirected graph $\mathcal{G} = (\mathcal{V}, \mathcal{E}, \mathbf{W})$, consisting of nodes $\mathcal{V} = \{1, ..., N\}$, edges $\mathcal{E} \subseteq \mathcal{V} \times \mathcal{V}$, and a weighted adjacency matrix $\mathbf{W} \in \mathbb{R}^{N \times N}$. Here $\mathbf{W}_{ij} = 0$ if $(i, j) \notin \mathcal{E}$. Define the *graph Laplacian* $\mathbf{L} = \mathbf{D} - \mathbf{W}$, with $\mathbf{D} = \text{diag}\left(\sum_{j=1}^N \mathbf{W}_{ij}\right)$. The *normalised* graph Laplacian is $\tilde{\mathbf{L}} := \mathbf{D}^{-1/2}\mathbf{L}\mathbf{D}^{-1/2}$, whose spectrum lies in $[0, 2]$ (Chung, 1997).

**Graph node kernels**. A graph node kernel is a symmetric, positive semidefinite function $k : \mathcal{V} \times \mathcal{V} \to \mathbb{R}$, mapping pairs of graph nodes to real numbers (Smola and Kondor, 2003). Heuristically, it assigns similarity scores to every pair of nodes, which are assembled into a *Gram matrix* $\mathbf{K} := [k(i, j)]_{i,j=1}^N \in \mathbb{R}^{N \times N}$. Many popular graph node kernels are parameterised as functions of $\mathbf{W}$, $\mathbf{L}$ or $\tilde{\mathbf{L}}$, expressed using the power series

$$\mathbf{K}_\alpha(\mathbf{W}) = \sum_{r=0}^\infty \alpha_r \mathbf{W}^r, \quad \alpha_r \in \mathbb{R} \ \ \forall r \in \{0, 1, ..., \infty\}. \tag{1}$$

The coefficients $(\alpha_r)_{r=0}^\infty$ determine the behaviour of the kernel, e.g. whether it upweights long- or short-range interactions. For instance, the *graph diffusion kernel* $\mathbf{K}_{\text{diff}} := \exp(-\beta \mathbf{L})$ takes $\alpha_r = (-\beta)^r / r!$. Graph node kernels flexibly capture structural information about $\mathcal{G}$, providing a natural choice for the GP covariance (Borovitskiy et al., 2021; Reid et al., 2024a).

---

[2]Specifically, an NVIDIA GeForce RTX 2080 Ti GPU (11 GB memory).

**Gaussian processes**. Let us now consider modelling functions $h : \mathcal{V} \to \mathbb{R}$ defined on the graph nodes. A common task is to identify the node that maximises $h$, e.g. the most influential social media user, or 'patient zero' in an epidemiological contact network. We may wish to solve $x^* = \arg\max_{x \in \mathcal{V}} h(x)$. Suppose we have access to a sequence of $T$ noisy observations of the objective $y_t = h(x_t) + \varepsilon, \varepsilon \sim \mathcal{N}(0, \sigma_n^2), t \in (1, 2, ..., T)$ at distinct nodes $x_t \in \mathcal{V}$. $\sigma_n^2$ is the noise variance. A common choice of statistical surrogate for the objective function $h$, for which we can perform analytic Bayesian inference given observations $\mathcal{D}_T = \{(x_t, y_t)\}_{t=1}^T$ (also denoted $\mathcal{D}_T = \{\boldsymbol{x}, \boldsymbol{y}\}$), is the *Gaussian process* (GP) (Rasmussen, 2003):

$$h(x) \sim \mathrm{GP}(m(x), k(x, x')). \tag{2}$$

Here, $m(x)$ is the mean function and $k(x, x')$ the covariance function ('kernel'). In our setting, the input domain consists of the nodes of a fixed graph. We can 'train' the kernel parameters $(\alpha_r)_{r=0}^\infty$ by maximising the *log-marginal likelihood* on the training data $\mathcal{D}_T$, and then compute the analytic posterior mean and covariance:

$$m_{|\boldsymbol{y}}(x) = m(x) + k(x, \boldsymbol{x})[k(\boldsymbol{x}, \boldsymbol{x}) + \sigma_n^2 \mathbf{I}]^{-1}(\boldsymbol{y} - m(\boldsymbol{x})), \tag{3}$$

$$k_{|\boldsymbol{y}}(x, x') = k(x, x') - k(x, \boldsymbol{x})[k(\boldsymbol{x}, \boldsymbol{x}) + \sigma_n^2 \mathbf{I}]^{-1} k(\boldsymbol{x}, x'). \tag{4}$$

*Bayesian optimisation* (BO) (Jones et al., 1998; Močkus, 1974) uses the posterior mean and covariance – or related quantities, like samples from the posterior – to efficiently locate $x^*$ in the presence of uncertainty. BO trades off exploration and exploitation in a mathematically principled manner, helping us decide which nodes to query in our attempt to maximise $h$.

**Efficiency and scalability**. A core computational challenge with performing Bayesian inference on graphs using GPs is that even just *evaluating* a dense graph kernel $\mathbf{K}_\alpha$ generally incurs $\mathcal{O}(N^3)$ time complexity, let alone computing the matrix-vector products and matrix inversions that we will in general require for BO. This is because it involves computing functions like $\exp(\cdot)$ or $(\cdot)^{-1}$ of the $N \times N$ weighted adjacency matrix $\mathbf{W}$, which becomes expensive for big graphs. For Euclidean kernels, a common recourse to improve scalability is to use *random features* (Rahimi and Recht, 2007; Yang et al., 2014): stochastic, finite-dimensional features $\{\varphi(i)\}_{i=1}^N \in \mathbb{R}^m$ whose dot product is equal to the kernel evaluation in expectation, $k(i, j) = \mathbb{E}(\varphi(i)^\top \varphi(j))$. In close analogy for discrete domains, researchers recently introduced *graph random features* (GRFs) (Choromanski, 2023; Reid et al., 2023) – sparse random walk-based vectors for unbiased estimation of graph node kernels.

**Graph random features: sparse, sharp kernel estimators using random walks**. The mathematical details of GRFs are involved and can be safely omitted on a first reading, but their behaviour can be intuitively understood as follows. Consider a sequence of scalars $(f_l)_{l=0}^\infty$ satisfying $\sum_{l=0}^r f_l f_{r-l} = \alpha_r \; \forall \; r$, namely, the 'deconvolution' of $(\alpha_r)_{r=0}^\infty$. We refer to $f_l$ as the *modulation function*. Suppose that the power series $\boldsymbol{\Psi} := \sum_{l=0}^\infty f_l \mathbf{W}^l$ converges. Then it is straightforward to see that for symmetric $\mathbf{W}$ (undirected graphs), we have $\boldsymbol{\Psi}^\top \boldsymbol{\Psi} = \mathbf{K}_\alpha$. Powers of an adjacency matrix count walks on a graph: for instance, $\mathbf{W}_{ij}^l$ gives the (weighted) number of walks of length $l$ between nodes $i$ and $j$. Since $\mathbf{K}_\alpha$ converges, longer walks must eventually be discounted, either due to decaying $f_l$ or due to multiplication of edge weights that are less than 1. The key insight of GRFs is that we can compute a Monte Carlo estimate $\boldsymbol{\Phi} \in \mathbb{R}^{N \times N}$ that satisfies $\boldsymbol{\Psi} = \mathbb{E}(\boldsymbol{\Phi})$ by importance sampling random walks.

Concretely, we simulate random walks out of every node of the graph. Each random walk of length $L$ consists of a number of 'prefix subwalks' – namely, for each step $l < L$, the sequence of the first $l$ nodes visited. We keep track of 1) the weights of edges they traverse, 2) the modulation function $f$, and 3) their marginal probabilities. Using a simple formula, we can construct unbiased,[3] sparse $N$-dimensional vectors that satisfy $\mathbb{E}(\varphi(i)^\top \varphi(j)) =$

---

[3]It has been noted that the shared source of randomness actually introduces a $\mathcal{O}(1/n)$ bias term for estimates of diagonal kernel entries $[\mathbf{K}_\alpha]_{i,i}$. This is of little significance for large graphs with many walkers so, following convention (Choromanski, 2023; Reid et al., 2023), we omit further discussion. One *could* remove this bias by sampling two independent ensembles of random walks and taking $\hat{\mathbf{K}} = \boldsymbol{\Phi}_1 \boldsymbol{\Phi}_2^\top$, at the cost of losing the positive definiteness guarantee and thus (typically) worse performance.

$\mathbb{E}\left(\left[\mathbf{\Phi}\mathbf{\Phi}^\top\right]_{ij}\right) = [\mathbf{K}_\alpha]_{i,j}$. Alg. 1 below provides pseudocode. It is deliberately kept high-level for compactness; the interested reader can find more details in App. A.

---

**Algorithm 1**: Constructing a GRF vector $\varphi(i) \in \mathbb{R}^N$ to approximate $\mathbf{K}_\alpha(\mathbf{W})$

---

1   **Inputs**: Graph $\mathcal{G}$, modulation function $f : \mathbb{N} \to \mathbb{R}$, random walk sampler $p$.

2   **Output**: Set of sparse GRFs $\{\varphi(i)\}_{i=1}^N \in \mathbb{R}^N$ that satisfy $[\mathbf{K}_\alpha]_{i,j} = \mathbb{E}(\varphi(i)^\top \varphi(j))$.

3   **for** $i \in \mathcal{V}$:

4      *initialise* $\varphi(i) \leftarrow \mathbf{0}$

5      **for** `walker_idx` $\in 1, ..., n$:

6         *sample* `random_walk` $\sim p$

7         **for** `prefix_subwalk` $\in$ `random_walk`:

8            $\varphi(i)[$`prefix_subwalk`$[-1]] + = (\prod$ `traversed_edge_weights`$) *$ $f(\text{length}(\text{prefix\_subwalk}))/p(\text{prefix\_subwalk})$

9      *normalise* $\varphi(i)/ = n$

---

Remarkably, under mild assumptions on $\mathcal{G}$ and $(\alpha_r)_{r=0}^\infty$, GRFs provide very sharp estimates of $\mathbf{K}_\alpha$. In particular, the estimates satisfy exponential concentration bounds, whilst storing only $\mathcal{O}(1)$ nonzero entries per feature. See Theorem 1 for a formal statement. As we will see in Section 3, we can use the sparse kernel estimate $\hat{K} := \mathbf{\Phi}\mathbf{\Phi}^\top$ as an efficient alternative to the dense exact kernel $\mathbf{K}_\alpha$, speeding up inference from $\mathcal{O}(N^3)$ to $\mathcal{O}(N^{3/2})$.

## 3   SCALABLE POSTERIOR INFERENCE WITH GRFS

Next, we demonstrate how GRFs speed up inference. We begin by proving novel theoretical results (Section 3.1), and then describe our full efficient GP workflow (Section 3.2).

### 3.1   NOVEL THEORETICAL RESULTS

We first recall the following result for GRFs, proved by Reid et al. (2024b).

**Theorem 1. (GRFs are sparse and give sharp kernel estimates (Reid et al., 2024b)).** Consider a graph $\mathcal{G}$ with weighted adjacency matrix $\mathbf{W}$ and node degrees $\{d_i\}_{i=1}^N$. Suppose we sample GRFs $\{\varphi(i)\}_{i=1}^N$ by sampling $n$ random walks that terminate with probability $p$ at each timestep, with modulation function $f$. Suppose also that $c := \sum_{r=0}^\infty |f_r| \left(\max_{i,j \in [\![1,N]\!]} \mathbf{W}_{ij} d_i/(1-p)\right)^r$ is finite. Then we have that

$$\mathbb{P}\left(|\varphi(i)^\top \varphi(j) - [\mathbf{K}_\alpha]_{i,j}| > t\right) \leq 2\exp\left(-\frac{t^2 n^3}{2(2n-1)^2 c^4}\right). \tag{5}$$

Moreover, with probability at least $1 - \delta$, any GRF $\varphi(i)$ is guaranteed to be sparse, with at most $n \log\left(1 - (1-\delta)^{1/n}\right)\log(1-p)^{-1}$ nonzero entries.

*Proof.* The proof, based on McDiarmid's inequality, is reported by Reid et al. (2024b). ∎

Theorem 1 demonstrates that, despite being sparse, GRFs give sharp kernel estimates. In particular, we can use Eq. (5) to compute the number of walkers $n$ needed to guarantee an accurate estimate of $\mathbf{K}_\alpha$ with high probability. Because of the bound, this number is *independent of the graph size $N$*. $n$ then determines the number of nonzero entries in the GRF, which also inherits independence of graph size $N$. We note that Theorem 1 makes the assumption about the graph $\mathcal{G}$ that the constant $c$ is finite. This is not controversial; Reid et al. (2024b) provide extensive discussion. Intuitively, it is natural that the spectrum of $\mathbf{W}$ must lie in some radius of convergence in order for the power series $\sum_{r=0}^\infty \alpha_r \mathbf{W}^r$ to converge. The condition for its Monte Carlo estimate to converge is only slightly stronger.

For computational reasons we often only sample random walks up to some fixed maximum length $l_{\max}$, e.g. a fraction of the graph diameter, whereupon $f_l = 0 \ \forall \ l > l_{\max}$ (discussed in App. C.1). The condition thus trivially holds in any reasonable implementation.

Given Theorem 1, we will henceforth assume that the number of walkers $n$ is constant, confident that this gives a sharp kernel estimate. Property (2) of Theorem 2 is novel.

**Theorem 2. (Properties of $\hat{\mathbf{K}}$)**. The randomised approximate Gram matrix $\hat{\mathbf{K}} := \mathbf{\Phi}\mathbf{\Phi}^\top = \left[\varphi(i)^\top\varphi(j)\right]_{i,j=1}^N \in \mathbb{R}^{N \times N}$ has the following properties.

1. *Property 1.* $\hat{\mathbf{K}}$ supports $\mathcal{O}(N)$ matrix-vector multiplication;

2. *Property 2.* The condition number of the approximate Gram matrix $\kappa\left(\hat{\mathbf{K}} + \sigma_n^2\mathbf{I}\right)$ is $\mathcal{O}(N)$.

*Proof.* Property (1) follows trivially from the fact that $\hat{\mathbf{K}}$ has $\mathcal{O}(N)$ nonzero entries, whereupon matrix-vector multiplication only requires $\mathcal{O}(N)$ operations. Considering (2), since $\hat{\mathbf{K}}$ is positive definite, the smallest possible eigenvalue of $\hat{\mathbf{K}} + \sigma_n^2\mathbf{I}$ is $\sigma_n^2$. Then note that

$$\|\hat{\mathbf{K}}\|_2 \leq \|\hat{\mathbf{K}}\|_{\mathrm{F}} := \sqrt{\sum_{i,j=1}^N |\hat{\mathbf{K}}_{i,j}|^2} = \sqrt{\sum_{i,j=1}^N |\varphi(i)^\top\varphi(j)|^2} \leq N \max_{i,j}|\varphi(i)^\top\varphi(j)|. \tag{6}$$

Under the assumptions above $\|\varphi(i)\|_1 \leq c \ \forall \ i$, whereupon $|\varphi(i)^\top\varphi(j)| \leq c^2 \ \forall \ i,j$. Hence, we have that $\kappa\left(\hat{\mathbf{K}} + \sigma_n^2\mathbf{I}\right) \leq 1 + N\frac{c^2}{\sigma_n^2}$, which is $\mathcal{O}(N)$ as claimed. ∎

Theorem 2 immediately implies the following corollary, which is also novel for GRFs.

**Lemma 1. Solving the sparse linear system.** Consider solving $\left(\hat{\mathbf{K}} + \sigma_n^2\mathbf{I}\right)\boldsymbol{v} = \boldsymbol{b}$, where $\boldsymbol{v}, \boldsymbol{b} \in \mathbb{R}^N$. This can be achieved with the conjugate gradient method in $\mathcal{O}\left(N^{3/2}\right)$ time.

*Proof.* Using the conjugate gradient method, it is known that the system can be solved in $\sqrt{\kappa\left(\hat{\mathbf{K}} + \sigma_n^2\mathbf{I}\right)}$ iterations[4] (Shewchuk, 1994), which by property (2) above is $\mathcal{O}\left(N^{1/2}\right)$. Each iteration involves matrix-vector multiplication, which is $\mathcal{O}(N)$ in our case due to property (1). Combining gives a total time complexity of $\mathcal{O}\left(N^{3/2}\right)$. ∎

We remark that this is substantially less than the $\mathcal{O}\left(N^3\right)$ time complexity of exact GP methods that use $\mathbf{K}_\alpha$ rather than $\hat{\mathbf{K}}$. It is also straightforward to see that Theorem 2 and Lemma 1 will continue to hold if we only consider a subset of the nodes of the graph, e.g. just considering a set of training nodes of cardinality $N_{\mathrm{train}} \leq N$.

## 3.2 From pathwise conditioning to conjugate gradients

We now introduce the three-step 'recipe' of posterior inference using GRFs: *kernel initialisation*, *hyperparameter learning* and *posterior inference*. We will also analyse the overall time and space complexity of this workflow. App. C.1 gives further heuristic guidance for practitioners, including for choosing the number of walkers $n$.

**Kernel initialisation**. We compute the Gram matrix using Alg. 1, which involves sampling $n$ random walks for every node on the graph. This yields a sparse kernel approximation:

$$\hat{\mathbf{K}} := \mathbf{\Phi}\mathbf{\Phi}^\top = \left[\varphi(i)^\top\varphi(j)\right]_{i,j=1}^N \in \mathbb{R}^{N \times N}. \tag{7}$$

In practice, $\hat{\mathbf{K}}$ does not need to be materialised as we can replace the matrix-vector product $\hat{\mathbf{K}}\boldsymbol{v}$ with two fast matrix-vector products $\mathbf{\Phi}(\mathbf{\Phi}^\top\boldsymbol{v})$. Each is computed in linear time.

**Hyperparameter learning**. Denote the training data $\mathcal{D}_T = \{\boldsymbol{x}, \boldsymbol{y}\}$, containing training nodes $\boldsymbol{x}$ and corresponding noisy observations $\boldsymbol{y}$. We learn the hyperparameters $\boldsymbol{\theta}$, such as observation noise and the modulation function $f$, by maximising the log marginal likelihood,

$$\mathcal{L}(\boldsymbol{\theta}) = -\frac{1}{2}\boldsymbol{y}^\top\mathbf{H}_{\boldsymbol{\theta}}^{-1}\boldsymbol{y} - \frac{1}{2}\log\det(\mathbf{H}_{\boldsymbol{\theta}}) - \frac{N}{2}\log(2\pi), \tag{8}$$

---

[4]Following prior work, we use 'solving' as shorthand for reducing the error on our approximation of $\boldsymbol{v}$ by a factor of $\varepsilon$ (Maddox et al., 2021), here set to $10^{-2}$. We will see that this is empirically robust.

where $\mathbf{H_\theta} = \left(\hat{\mathbf{K}}_{\boldsymbol{xx}} + \sigma_n^2 \mathbf{I}\right)$. We use the Adam optimiser and estimate the gradient,

$$\nabla \mathcal{L}(\boldsymbol{\theta}) = \frac{1}{2}\left(\mathbf{H_\theta^{-1}}\boldsymbol{y}\right)^\top \frac{\partial \mathbf{H_\theta}}{\partial \boldsymbol{\theta}}\left(\mathbf{H_\theta^{-1}}\boldsymbol{y}\right) - \frac{1}{2}\,\mathrm{tr}\left(\mathbf{H_\theta^{-1}}\frac{\partial \mathbf{H_\theta}}{\partial \boldsymbol{\theta}}\right), \tag{9}$$

using iterative methods (Gardner et al., 2018; Lin et al., 2024a). These avoid explicit matrix inverses via iterative linear system solvers such as conjugate gradients (CGs) (Hestenes and Stiefel, 1952; Shewchuk, 1994). Since CGs rely on matrix-vector multiplication, this allows us to leverage the efficient structure of GRFs. Meanwhile, the trace term is estimated using Hutchinson's trace estimator (Hutchinson, 1990),

$$\mathrm{tr}\left(\mathbf{H_\theta^{-1}}\frac{\partial \mathbf{H_\theta}}{\partial \boldsymbol{\theta}}\right) = \mathbb{E}\left(\boldsymbol{z}^\top \mathbf{H_\theta^{-1}}\frac{\partial \mathbf{H_\theta}}{\partial \boldsymbol{\theta}}\boldsymbol{z}\right) \approx \frac{1}{S}\sum_{s=1}^{S}\boldsymbol{z}_s^\top \mathbf{H_\theta^{-1}}\frac{\partial \mathbf{H_\theta}}{\partial \boldsymbol{\theta}}\boldsymbol{z}_s, \tag{10}$$

where $\boldsymbol{z}_s$ are random probes satisfying $\mathbb{E}\left[\boldsymbol{z}_s\boldsymbol{z}_s^\top\right] = \mathbf{I}$. This gives a batch of linear systems,

$$\mathbf{H_\theta}\left[\boldsymbol{v_y}, \boldsymbol{v}_1, ..., \boldsymbol{v}_S\right] = [\boldsymbol{y}, \boldsymbol{z}_1, ..., \boldsymbol{z}_S], \tag{11}$$

which can be solved via iterative methods. The solutions allow us to estimate $\nabla \mathcal{L}$.

**Posterior inference**. We perform posterior inference using pathwise conditioning (Wilson et al., 2020; 2021) and iterative methods – a combination that has attracted recent interest in the literature (Lin et al., 2024b; 2025). This allows us to exploit the efficient structure of GRFs. In particular, pathwise conditioning expresses a sample from the posterior as a sample from the prior with an additional correction term,

$$\boldsymbol{g}_{|\boldsymbol{y}}(\cdot) = \boldsymbol{g}(\cdot) + \hat{\mathbf{K}}_{(\cdot)\boldsymbol{x}}\left(\hat{\mathbf{K}}_{\boldsymbol{xx}} + \sigma_n^2 \mathbf{I}\right)^{-1}(\boldsymbol{y} - (\boldsymbol{g}(\boldsymbol{x}) + \boldsymbol{\varepsilon})), \tag{12}$$

where $(\cdot)$ is any node of the graph $\mathcal{G}$, $\boldsymbol{g}_{|\boldsymbol{y}}$ is a sample from the posterior, $\boldsymbol{g}$ is a sample from the prior, and $\boldsymbol{\varepsilon} \sim \mathcal{N}\left(\mathbf{0}, \sigma_n^2 \mathbf{I}\right)$. This facilitates the use of iterative linear system solvers to compute $\left(\hat{\mathbf{K}}_{\boldsymbol{xx}} + \sigma_n^2 \mathbf{I}\right)^{-1}(\boldsymbol{y} - (\boldsymbol{g}(\boldsymbol{x}) + \boldsymbol{\varepsilon}))$, which again avoids the explicit inverse and leverages sparse matrix multiplication.[5] Once more, we use CGs (Hestenes and Stiefel, 1952; Shewchuk, 1994) as linear system solver, though alternatives have recently been proposed (Lin et al., 2023; 2024c). The structure of the GRFs kernel also admits efficient sampling from the prior via $\boldsymbol{g} = \boldsymbol{\Phi}\boldsymbol{w}$ with $\boldsymbol{w} \sim \mathcal{N}(\mathbf{0}, \mathbf{I})$,[6] since $\mathrm{Cov}(\boldsymbol{\Phi}\boldsymbol{w}) = \boldsymbol{\Phi}\boldsymbol{\Phi}^\top = \hat{\mathbf{K}}$.

**Algorithm complexity**. Kernel initialisation takes $\mathcal{O}(N)$ time, since a fixed number of random walks are simulated from all $N$ nodes. Training and inference are dominated by CG solvers, with $\mathcal{O}\left(N^{3/2}\right)$ time complexity (Lemma 1). All stages use sparse matrices (e.g. $\hat{\mathbf{K}} + \sigma_n^2 \mathbf{I}$) with $\mathcal{O}(N)$ nonzero entries, giving overall space complexity $\mathcal{O}(N)$.

## 4 EXPERIMENTAL RESULTS

Here, we present empirical results demonstrating the scalability and practical effectiveness of the GRF-GPs model. In each case, full experimental details are provided in App. C.

### 4.1 COMPUTATION COMPLEXITY AND ABLATIONS

**Dense vs. sparse GRFs: the importance of an efficient implementation**. We benchmark posterior inference on synthetic graphs under two GRF implementations. First, we consider a dense baseline that uses GRFs, but explicitly materialises the $N \times N$ kernel approximation and computes its inverse. Second, we take the sparse GRF method described

---

[5]An alternative to solving this sparse linear system is to use the Johnson-Lindenstrauss transformation to reduce the dimensionality of the features $\{\varphi(i)\}_{i=1}^N \in \mathbb{R}^N$, whilst preserving their dot products in expectation (Dasgupta and Gupta, 2003). At the cost of sacrificing sparsity, we can then use the Woodbury Identity to efficiently solve a smaller linear system. We describe this in App. B.

[6]$\boldsymbol{g}$ is an $N$-dimensional vector corresponding to a sample evaluated at all $N$ nodes. One could consider a subset of nodes, where the prior sample $\boldsymbol{g}(\cdot)$ now corresponds to the vector's $(\cdot)$th entry.

in Section 3.2, storing the random walk trajectories and solving the corresponding linear systems with CG methods. Table 1 summarises the results, with full measurements provided in the App. C.2 (Table 2 and Table 3). For a graph with 8192 nodes, we observe a **50×** **speedup** in total wall-clock time.

Table 1: **GRF-GPs have sub-quadratic time scaling and linear memory scaling**. Empirical scaling exponents ($\pm$ s.d.) for memory usage, kernel initialisation, training, and inference with respect to graph size $N$. In the table, an entry $b$ indicates scaling $\mathcal{O}(N^b)$.

| Kernel | Memory | Kernel init. time | Training time | Inference time |
|---|---|---|---|---|
| GRFs (Dense) | $2.00 \pm 0.00$ | $1.21 \pm 0.06$ | $1.97 \pm 0.38$ | $2.16 \pm 0.33$ |
| GRFs (Sparse) | $1.00 \pm 0.00$ | $0.81 \pm 0.04$ | $1.04 \pm 0.04$ | $1.04 \pm 0.05$ |

Figure 2 shows log–log scaling curves. Exponents from the asymptotic regime match those shown in Table 1. As expected, GRFs attain **linear** memory and initialisation cost, and **sub-quadratic** training and inference, scaling to graphs with **1M nodes**. The near-linear runtime trends in training and inference reflect the fixed iteration budget of sparse linear solves; conditioning effects have not yet dominated at these scales.

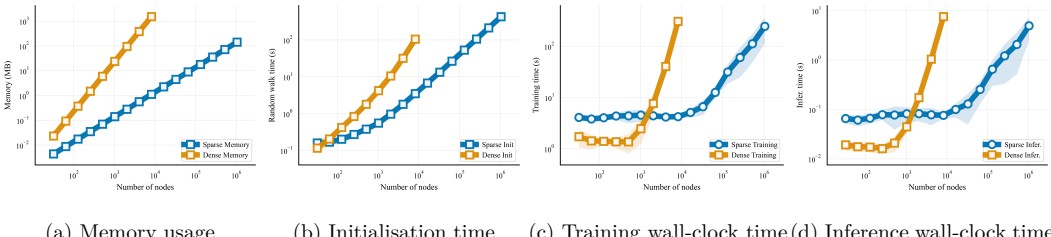

(a) Memory usage     (b) Initialisation time     (c) Training wall-clock time (d) Inference wall-clock time

Figure 2: **GRFs scale better (blue curve) when sparsity is leveraged**. Scaling experiments for the GRF-GPs. Yellow: brute-force dense implementation. Blue: sparse implementation. Panels (a)–(d) correspond to memory footprint, kernel initialisation time, training time and inference time, respectively. The dense model is limited to 8,192 nodes due to its higher memory demands.

**Importance sampling ablation.** As discussed in Section 2, the key insight of GRFs is that one can replace a function of a weighted adjacency matrix $\mathbf{W}$ with a Monte Carlo estimate. This estimate is constructed using random walks, weighted by (1) the product of traversed edge weights and (2) the per-walk probability under the sampling mechanism. Following Reid et al. (2024b), one can investigate the significance of this principled approach by instead constructing a naive random walk-based empirical kernel, without appropriate reweighting. In particular, we replace line 8 of Alg 1 by

$$\varphi(i)[\texttt{prefix\_subwalk}[-1]] + = \left(\prod \texttt{traversed\_edge\_weights}\right) * f(\texttt{length}(\texttt{prefix\_subwalk})), \quad (13)$$

removing normalisation by $p(\texttt{prefix\_subwalk})$. Crucially, this set of features *still* defines a valid kernel on $\mathcal{G}$, but it is no longer an unbiased estimate of a power series of $\mathbf{W}$. A similar 'ad-hoc' kernel was used in the context of transformer position encodings by Choromanski et al. (2022). Full empirical results are reported in App. C.3, where we find this modification substantially degrades regression performance. Intuitively, failing to upweight long, unlikely walks by $1/p(\texttt{prefix\_subwalk})$ makes it challenging to model longer-range dependencies.

## 4.2 REGRESSION TASKS

Next, we apply our method to regression with a variety of real-world datasets.

**1. Traffic speed prediction**. To assess predictive capability, we begin with a traffic speed forecasting task (Figure 7) on the San Jose freeway sensor network (Chen et al., 2001). We follow the setup of Borovitskiy et al. (2021). Experiment details can be found in App. C.5.

We compare three kernel configurations by measuring the negative log probability density (NLPD) and the root mean squared error (RMSE) of the maximum-a-posteriori (MAP) predictions. We consider (1) the exact diffusion kernel $\mathbf{K}_{\text{diff}}$; (2) a GRF kernel in a 'diffusion shape' (namely, the modulation function frozen to approximate $\mathbf{K}_{\text{diff}}$, with a learnable lengthscale); and (3) a GRF kernel with a flexible, fully learnable modulation function.

Figure 3 (a)-(b) reports the test NLPD and RMSE as a function of the number of random walks per node $n$. As $n$ increases, the variance of the Monte Carlo approximation $\hat{\mathbf{K}}$ drops. It better captures the underlying graph structure, yielding more accurate predictions. Note that the fully-learnable GRF kernel consistently outperforms the diffusion-shaped variant, highlighting the benefit of implicit kernel learning via a flexible modulation function.

In addition to greater flexibility via learnable $f_l$, another reason GRFs are able to outperform $\mathbf{K}_{\text{diff}}$ may be that their inbuilt sparsity is actually a sensible inductive bias. Pairs of graph nodes only have nonzero covariance if their respective ensembles of random walks *hit*, which is more likely if they are nearby in $\mathcal{G}$. This means that a node's predictions depend mostly on information from its local neighborhood, whilst still sampling longer dependencies with lower probability. In contrast, dense kernels can sometimes be prone to the 'oversmoothing' effect as they capture spurious long-range correlations driven by noise (Keriven, 2022).

**2. Wind interpolation on the globe**. Next, we consider the task of interpolating monthly average wind velocities from the ERA5 dataset (Hersbach et al., 2019), from a set of locations on the Aeolus satellite track (Reitebuch, 2012). Our problem setup follows that of Wyrwal et al. (2024) and Robert-Nicoud et al. (2023). We discretise the surface of the globe (formally, the manifold $S^2$) by computing a $k$-nearest neighbours graph from the observation locations. This yields a graph $\mathcal{G}$ with 10K nodes, with which we can apply our scalable GRF-GPs algorithm. The task is to predict the velocity fields of a held out test set.

The test NLPD and RMSE of the diffusion-shape and fully-learnable GRF kernels are shown in Figure 3 (c)-(d). Similarly, the predictions improve as $n$ increases. We provide full results and visualisations in App. C.6. This type of implicit manifold GP regression – approximating a (possibly unknown) manifold by computing a nearest neighbour graph $\mathcal{G}$ and then performing inference therein – is a rich area of active research (Borovitskiy et al., 2021; Dunson et al., 2021; Fichera et al., 2023). This is an exciting possible application of GRFs; we hope our initial example will spur future work.

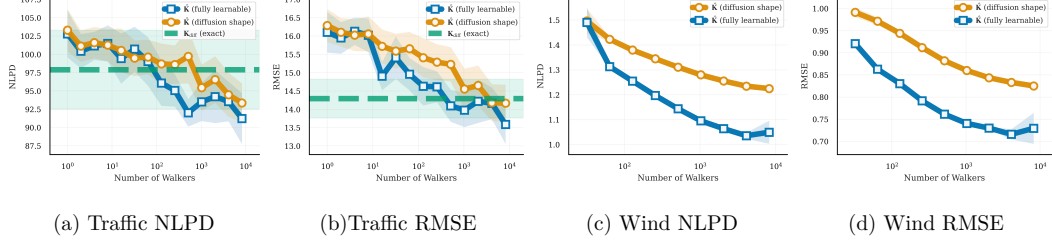

(a) Traffic NLPD     (b)Traffic RMSE     (c) Wind NLPD     (d) Wind RMSE

Figure 3: **GRFs outperforms diffusion baselines in regression tasks.** Panels (a)–(d) report test NLPD and RMSE versus the number of random walkers $n$. Blue: GRF kernel with a fully learnable modulation; orange: diffusion-shape GRF. Shading shows $\pm 1$ s.d. On **Traffic**, the learnable GRF surpasses the exact diffusion kernel once $n \gtrsim 500$. On **Wind**, the exact diffusion kernel is omitted due to $\mathcal{O}(N^3)$ cost. Again, the fully-learnable GRF kernel consistently achieves lower NLPD and RMSE than the diffusion-shape variant.

### 4.3 Scalable and robust Bayesian optimisation

Having demonstrated the scalability of GRFs (Section 4.1) and their efficacy for GP regression (Section 4.2), we now use them to perform efficient Bayesian optimisation (BO). We consider large graphs with up to $10^6$ nodes, where exact posterior inference becomes prohibitively expensive. For the acquisition strategy we use *Thompson sampling*, drawing samples from the posterior over the objective function and selecting maximisers as the next

query point (Russo et al., 2018; Thompson, 1933). Posterior sampling is made efficient by pathwise conditioning, given in Equation (12). Alg. 3 in App. C.7 gives full details.

**Datasets and baselines**. For datasets, we consider a range of synthetic and real-world graphs. First, we maximise a variety of scalar functions on grids, community and circular graphs, chosen to have different properties, e.g. multimodality and periodicity. Next, we identify 'influential' (high node degree) users in a range of social networks: Enron, Facebook, Twitch and YouTube. Lastly, we predict the physical location with the greatest windspeed for the ERA5 dataset studied in Section 4.2, considering three different altitudes where the wind behaviour is known to be qualitatively different (Wyrwal et al., 2024). In each case, we compare our efficient BO method with random search, breadth first search and depth first search policies. In almost all instances, our algorithm achieves lower regret, showing the benefit of uncertainty-aware strategies for large-scale optimisation on graphs.

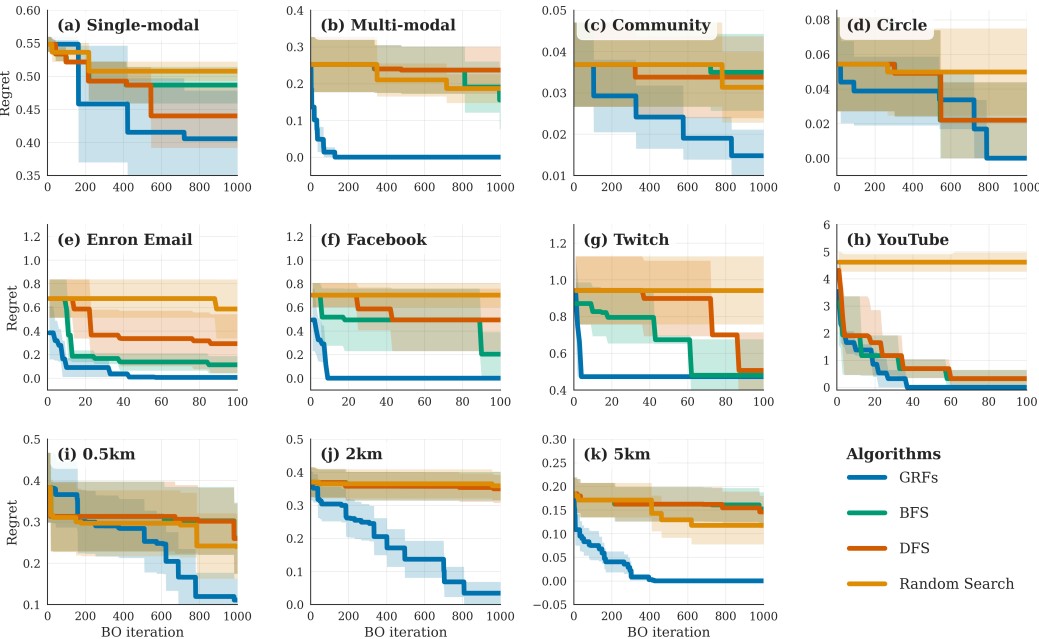

Figure 4: **GRF-based BO achieves lower regret than search-based baselines in most datasets.** Each panel shows the regret curve of BO for the following datasets: (a)-(d) synthetic datasets, (e)-(h) social networks, and (i)-(k) windspeed in the ERA5 dataset.

### 4.4 FUTURE WORK: SCALABLE VARIATIONAL GPS FOR CLASSIFICATION

Lastly, we evaluate GRF-GPs on a multi-class node classification task using the Cora citation network benchmark (McCallum et al., 2000). In this non-conjugate inference setting, we handle the non-Gaussian likelihood via variational inference (Leibfried et al., 2020). Pathwise conditioning for classification is nontrivial (Wilson et al., 2021); we defer a full treatment to a future paper. We can nonetheless assess the performance of GRFs, even without explicit time complexity guarantees like Lemma 1. Details are provided in App. C.8. Once again, sparse GRF kernels achieve very strong performance.

## 5 CONCLUSION

We demonstrated how graph random features (GRFs), a recently-introduced Monte Carlo algorithm, can be used to speed up training and inference with Gaussian processes on discrete input spaces. Under mild assumptions, GRFs support $\mathcal{O}(N^{3/2})$ time complexity inference – much faster than $\mathcal{O}(N^3)$ for their exact counterpart – with probabilistic accuracy guarantees. This translates to substantial wall-clock time speedups, and unlocks scalable Bayesian optimisation on massive topologies with little or no sacrifice in performance.

## 6 Ethics and reproducibility

**Ethics.** Our work is methodological and does not raise direct ethical concerns. Nonetheless, advances in scalable graph-based ML may amplify risks if misapplied, either by malicious actors or through unforeseen downstream consequences.

**Reproducibility.** All datasets are freely available online, with links to the original sources provided. The code publicly available at https://github.com/MatthewZhang473/Graph-Random-Features-for-Scalable-Gaussian-Processes.

**Trustworthy, scalable and robust machine learning for graphs**. Much of graph-based machine learning research to date has focused on graph neural networks. Kernel methods provide an alternative which is inherently more explainable. Our algorithms are applicable to a broad range of real world settings, including many tasks in social networks. We demonstrate strong scalability, efficiency and performance whilst retaining the benefits of explainability.

## 7 Funding and acknowledgements

JAL is supported by the University of Cambridge Harding Distinguished Postgraduate Scholars Programme. AW acknowledges support from EPSRC via a Turing AI Fellowship under grant EP/V025279/1 and project FAIR under grant EP/V056883/1. AW is also supported by the Leverhulme Trust via CFI. RET is supported by the EPSRC Probabilistic AI Hub (EP/Y028783/1). IR gratefully acknowledges funding from Trinity College, Cambridge and from Google. IR was a student researcher at Google DeepMind at the time of submission. This work was funded in part by an unrestricted gift from Huawei.

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

## A  FULL GRF ALGORITHM

To complement the pseudocode provided in Alg. 1, Alg. 2 provides a more detailed explanation of how one can estimate graph node kernels using graph random features (GRFs). The motivated reader is invited to consult the works of Reid et al. (2023) and Choromanski (2023) for the original accounts, including further intuitions and a proof of unbiasedness.

---

**Algorithm 2**: Constructing a random feature vector $\varphi(i) \in \mathbb{R}^N$ to approximate $\mathbf{K}_\alpha(\mathbf{W})$

---

    **Inputs**: weighted adjacency matrix $\mathbf{W} \in \mathbb{R}^{N \times N}$ for a graph $\mathcal{G}$ with $N$ nodes, vector of
1  unweighted node degrees $\boldsymbol{d} \in \mathbb{R}^N$, modulation function $f : (\mathbb{N} \cup \{0\}) \to \mathbb{R}$, termination probability $p_{\text{halt}} \in (0, 1)$, node $i \in \mathcal{N}$, number of random walks to sample $n \in \mathbb{N}$.

2  **Output**: random walk feature vector $\varphi(i) \in \mathbb{R}^N$.

3  initialise: $\varphi(i) \leftarrow \mathbf{0}$

4  **for** $w = 1, ..., n$

---

**Algorithm 2**: Constructing a random feature vector $\varphi(i) \in \mathbb{R}^N$ to approximate $\mathbf{K}_\alpha(\mathbf{W})$

---

5     initialise: `load` $\leftarrow 1$

6     initialise: `current_node` $\leftarrow i$

7     initialise: `terminated` $\leftarrow$ False

8     initialise: `walk_length` $\leftarrow 0$

9     **while** `terminated` $=$ False **do**

10      $\varphi(i)\,[\texttt{current\_node}] \leftarrow \varphi(i)\,[\texttt{current\_node}] + \texttt{load} \times f\,(\texttt{walk\_length})$

11      `walk_length` $=$ `walk_length`$+1$

12      `new_node` $\leftarrow \mathrm{Unif}[\mathcal{N}(\texttt{current\_node})]$          ▷ assign to one of neighbours

13      `load` $\leftarrow$ `load` $\times \frac{d[\texttt{current\_node}]}{1-p_{\mathrm{halt}}} \times \mathbf{W}[\texttt{current\_node}, \texttt{new\_node}]$    ▷ update load

14      `current_node` $\leftarrow$ `new_node`

15      `terminated` $\leftarrow (\mathrm{t} \sim \mathrm{Unif}(0,1) < p_{\mathrm{halt}})$     ▷ draw RV to decide on termination

16    **end while**

17 **end for**

18    normalise $\varphi(i) = \varphi(i)/n$

---

# B   Efficiently solving linear systems $(\hat{\mathbf{K}} + \sigma_n^2 \mathbf{I})\boldsymbol{v} = \boldsymbol{b}$ with the Woodbury Formula

In this appendix, we provide another algorithm for efficiently solving linear system $(\hat{\mathbf{K}} + \sigma_n^2 \mathbf{I})\boldsymbol{v} = \boldsymbol{b}$ , with the use of *Woodbury matrix identity formula* and Johnson-Lindenstrauss Transform (JLT) (Freksen, 2021). This algorithm has time complexity $O(N^2 m + m^3)$, where $m \ll N$ is the number of the output dimensions (a hyperparameter of the JLT algorithm). While this approach appears promising, we emphasise that our investigation here is preliminary. A more thorough evaluation of its empirical performance and potential trade-offs is left to future work.

Take the decomposition of $\hat{\mathbf{K}}$ of the form $\hat{\mathbf{K}} = \boldsymbol{\Phi}\boldsymbol{\Phi}^\top$. Construct a random Gaussian matrix $\mathbf{G} \in \mathbb{R}^{N \times m}$, with entries taken independently at random from the Gaussian distribution with mean $\mu = 0$ and standard deviation $\sigma = 1$. By the JLT, we can unbiasedly approximate $\boldsymbol{\Phi}\boldsymbol{\Phi}^\top$ as $\mathbf{K}_1 \mathbf{K}_1^\top$, where $\mathbf{K}_1 = \frac{1}{\sqrt{m}}\boldsymbol{\Phi}\mathbf{G} \in \mathbb{R}^{N \times m}$ (and with strong concentration guarantees for $m$ of logarithmic in $N$ order). We can then approximately rewrite $(\hat{\mathbf{K}} + \sigma_n^2 \mathbf{I})^{-1}\boldsymbol{b}$ as $\frac{1}{\sigma_n^2}(\mathbf{I}_N + \mathbf{U}\mathbf{U}^\top)^{-1}\boldsymbol{b}$, for $\mathbf{U} = \frac{\mathbf{K}_1}{\sigma_n}$.

Now, we can apply the following special case of the celebrated Woodbury Matrix Identity formula:

$$(\mathbf{I}_N + \mathbf{U}\mathbf{U}^\top)^{-1} = \mathbf{I}_N - \mathbf{U}(\boldsymbol{I}_m + \mathbf{U}^\top\mathbf{U})^{-1}\mathbf{U}^\top. \tag{14}$$

Therefore, we conclude that the solution $\boldsymbol{v}$ to our linear system can be approximated as:

$$\boldsymbol{v} \approx \left[\mathbf{I}_N - \mathbf{U}(\boldsymbol{I}_m + \mathbf{U}^\top\mathbf{U})^{-1}\mathbf{U}^\top\right]\boldsymbol{b}. \tag{15}$$

The expression on the right side can clearly be computed in time $O(Nm + m^3)$ since brute-force inversion of $\mathbf{X} = (\boldsymbol{I}_m + \mathbf{U}^\top\mathbf{U})^{-1} \in \mathbb{R}^{m \times m}$ takes $O(m^3)$ time and expression $\mathbf{U}(\mathbf{X}\mathbf{U}^\top\boldsymbol{b})$ for $\mathbf{U} \in \mathbb{R}^{N \times m}$ can be computed in $O(Nm)$ time. Thus, since the computation of $\mathbf{K}_1$ takes time $O(N^2 m)$, total time complexity is $O(N^2 m + m^3)$.

This approach, using dimensionality reduction of GRFs to replace the inverse of an $N \times N$ matrix by the inverse of an $m \times m$ matrix, provides an interesting alternative to relying on sparse operations to achieve speedups.

## C   EXPERIMENT DETAILS

We provide experimental details in this section. All experiments are conducted on a single compute node equipped with an NVIDIA GeForce RTX 2080 Ti GPU (11 GB memory).

### C.1   CHOOSING GRF HYPERPARAMETERS: GUIDANCE FOR PRACTITIONERS

In this appendix, we provide further practical guidance for practitioners when choosing the number of random walks $n$ and (if desired) the maximum walk length $l_{\max}$.

**Choosing $n$**. Theorem 1 gives a precise formula for choosing the number of walkers $n$ to guarantee an accurate kernel estimate with high probability. In principle, one could use this to derive the minimum $n$ required for a sharp estimate, given the constant $c$, the maximum permissible deviation $t$, and the maximum permissible probability of deviation $\mathbb{P}\left(|\varphi(i)^\top \varphi(j) - [\mathbf{K}_\alpha]_{i,j}| > t\right)$. However, in practice we find this to be unnecessary: choosing $n$ to be a small multiple of the average node degree already works well. As seen in Figure 3, performance tends to improve as $n$ increases, at the cost of decreasing kernel sparsity and thus slower wall-clock times. We recommend choosing $n$ that balances the practitioner's performance and efficiency requirements.

**Choosing $l_{\max}$**. In Section 3.1, we noted that in implementations it is often convenient and memory-efficient to only sample walks up to some maximum length $l_{\max}$. This way, the number of modulation function terms $(f_l)_{l=0}^{l_{\max}}$ that must be learned is finite and fixed. We emphasise that this is *not a requirement for the time complexity guarantees in Section 3.1*; it is a practical (as opposed to mathematical) detail. In principle, one could choose $l_{\max}$ to be sufficiently large that all $n$ walkers will be shorter with high probability, avoiding any truncation – see e.g. App. A.1 by Reid et al. (2023) for a mathematical bound.[7] However, in practice we find that choosing $l_{\max}$ to be some modest fraction of the graph diameter is sufficient for good performance. In each experiment, we report $l_{\max}$ in the respective appendix.

### C.2   TIME AND SPACE COMPLEXITY MEASUREMENTS

This section reports experimental details for the scaling results in Figure 2 and Table 1.

**Synthetic data**. We generate synthetic signals on ring graphs of increasing size: $N = 2^5, 2^6, ..., 2^{20}$ nodes. The groundtruth functions are smooth periodic functions on the nodes with additive Gaussian noise ($\sigma_n^2 = 0.1$). For graphs with more than 8192 nodes, we only use the sparse GRF implementation, since the dense adjacency matrices exceed the available GPU memory. Random feature matrices $\mathbf{\Phi}$ are constructed using 100 random walks per node, with halting probability $p_{\text{halt}} = 0.1$. Walks longer than 3 hops are truncated.

**Measurements taken**. For each graph size, and across 5 random seeds, we measure:

- The **memory footprint** of the random feature matrices $\mathbf{\Phi}$.
- The **random-walk preprocessing time** for constructing $\mathbf{\Phi}$.
- **Training wall-clock time**, measured as total optimiser runtime over 50 epochs.
- **Inference wall-clock time**, measured as posterior mean and covariance evaluation time on the test set.

---

[7]As the number of walkers $n$ grows, the minimum value of $l_{\max}$ to ensure that all walkers are shorter than $l_{\max}$ with high probability scales logarithmically with $n$.

The dense implementation uses the GPflow library for kernels with explicit adjacency materialisation, while the sparse implementation uses a GPyTorch library to implement kernels with customised sparse linear operators to maximise efficiency (Gardner et al., 2018; Matthews et al., 2017; van der Wilk et al., 2020). Full empirical measurements are shown in Table 2 and Table 3.

Table 2: Memory and time measurements for dense implementation: mean $\pm$ s.d.

| Graph Size | Memory (MB) | Kernel init time (s) | Training time (s) | Inference time (s) |
|---|---|---|---|---|
| 32 | $0.024 \pm 0.000$ | $0.115 \pm 0.017$ | $1.726 \pm 0.336$ | $0.019 \pm 0.002$ |
| 64 | $0.094 \pm 0.000$ | $0.205 \pm 0.012$ | $1.430 \pm 0.219$ | $0.018 \pm 0.002$ |
| 128 | $0.375 \pm 0.000$ | $0.421 \pm 0.025$ | $1.403 \pm 0.116$ | $0.017 \pm 0.002$ |
| 256 | $1.500 \pm 0.000$ | $0.840 \pm 0.044$ | $1.371 \pm 0.152$ | $0.016 \pm 0.002$ |
| 512 | $6.000 \pm 0.000$ | $1.800 \pm 0.069$ | $1.370 \pm 0.288$ | $0.021 \pm 0.004$ |
| 1024 | $24.000 \pm 0.000$ | $4.189 \pm 0.204$ | $2.465 \pm 0.595$ | $0.045 \pm 0.006$ |
| 2048 | $96.000 \pm 0.000$ | $10.546 \pm 0.107$ | $7.680 \pm 1.649$ | $0.173 \pm 0.001$ |
| 4096 | $384.000 \pm 0.000$ | $31.749 \pm 1.246$ | $40.376 \pm 4.080$ | $1.043 \pm 0.006$ |
| 8192 | $1536.000 \pm 0.000$ | $104.839 \pm 2.026$ | $307.188 \pm 35.938$ | $7.572 \pm 0.000$ |

Table 3: Memory and time measurements for sparse implementation: mean $\pm$ s.d.

| Graph Size | Memory (MB) | Kernel init time (s) | Training time (s) | Inference time (s) |
|---|---|---|---|---|
| 32 | $0.004 \pm 0.000$ | $0.160 \pm 0.033$ | $4.103 \pm 0.216$ | $0.066 \pm 0.007$ |
| 64 | $0.008 \pm 0.000$ | $0.168 \pm 0.022$ | $3.823 \pm 0.136$ | $0.061 \pm 0.008$ |
| 128 | $0.015 \pm 0.000$ | $0.202 \pm 0.022$ | $4.036 \pm 0.191$ | $0.066 \pm 0.007$ |
| 256 | $0.030 \pm 0.000$ | $0.271 \pm 0.030$ | $4.369 \pm 0.349$ | $0.079 \pm 0.009$ |
| 512 | $0.059 \pm 0.000$ | $0.379 \pm 0.021$ | $4.395 \pm 0.619$ | $0.077 \pm 0.019$ |
| 1024 | $0.118 \pm 0.000$ | $0.552 \pm 0.024$ | $4.549 \pm 0.593$ | $0.082 \pm 0.014$ |
| 2048 | $0.235 \pm 0.000$ | $0.973 \pm 0.039$ | $4.416 \pm 0.320$ | $0.082 \pm 0.012$ |
| 4096 | $0.470 \pm 0.000$ | $1.790 \pm 0.028$ | $4.185 \pm 0.252$ | $0.078 \pm 0.015$ |
| 8192 | $0.938 \pm 0.000$ | $3.481 \pm 0.074$ | $4.247 \pm 0.143$ | $0.076 \pm 0.006$ |
| 16384 | $1.876 \pm 0.000$ | $6.764 \pm 0.052$ | $5.117 \pm 0.518$ | $0.100 \pm 0.016$ |
| 32768 | $3.751 \pm 0.000$ | $13.297 \pm 0.050$ | $6.623 \pm 1.048$ | $0.129 \pm 0.040$ |
| 65536 | $7.501 \pm 0.000$ | $26.569 \pm 0.063$ | $12.566 \pm 1.188$ | $0.254 \pm 0.061$ |
| 131072 | $15.001 \pm 0.000$ | $53.012 \pm 0.156$ | $31.534 \pm 6.376$ | $0.651 \pm 0.175$ |
| 262144 | $30.000 \pm 0.000$ | $105.901 \pm 0.514$ | $60.488 \pm 17.849$ | $1.216 \pm 0.443$ |
| 524288 | $60.000 \pm 0.000$ | $212.671 \pm 0.758$ | $111.672 \pm 31.377$ | $2.068 \pm 0.775$ |
| 1048576 | $120.000 \pm 0.000$ | $426.074 \pm 1.562$ | $245.060 \pm 65.159$ | $4.947 \pm 1.226$ |

**Scaling factor estimation**. We estimate empirical complexity exponents by fitting the measured runtime and memory data to a power-law model,

$$y \approx aN^b,$$

using ordinary least squares in log–log space, where $N$ is the number of graph nodes. Uncertainty in the slope $b$ is quantified with 95% confidence intervals derived from the $t$-

distribution. To capture asymptotic scaling behavior, fits are restricted to sufficiently large graphs: dense GP experiments are fit for $N \geq 2^9$, while sparse GP experiments are fit for $N \geq 2^{15}$. The fitted coefficients $a$ and $b$, together with confidence intervals and $R^2$ values, are summarised in Table 4.

Table 4: Fitted power-law scaling coefficients for memory usage, random-walk initialisation, training, and inference time. Each row reports multiplicative constant $a$, exponent $b$ with 95% confidence interval, and coefficient of determination $R^2$. Fits performed in log–log space.

|  | Kernel | $a$ | $b$ | 95% CI (b) | $R^2$ |
|---|---|---|---|---|---|
| Memory (MB) | Sparse | $1.37 \times 10^{-4}$ | 1.00 | $[1.00, 1.00]$ | 1.00 |
|  | Dense | $2.29 \times 10^{-5}$ | 2.00 | $[2.00, 2.00]$ | 1.00 |
| Kernel init time (s) | Sparse | $3.58 \times 10^{-3}$ | 0.81 | $[0.73, 0.88]$ | 0.97 |
|  | Dense | $1.22 \times 10^{-3}$ | 1.21 | $[1.09, 1.33]$ | 0.99 |
| Training time (s) | Sparse | $1.32 \times 10^{-4}$ | 1.04 | $[0.96, 1.12]$ | 1.00 |
|  | Dense | $3.93 \times 10^{-6}$ | 1.97 | $[1.20, 2.73]$ | 0.96 |
| Inference time (s) | Sparse | $2.79 \times 10^{-6}$ | 1.04 | $[0.93, 1.14]$ | 0.99 |
|  | Dense | $1.92 \times 10^{-8}$ | 2.16 | $[1.50, 2.81]$ | 0.97 |

## C.3 ABLATION STUDIES

This section reports the results of the ablation experiment described in Section 4.1, where we replace the GRF estimate of a function of a weighted adjacency matrix by an ad-hoc random walk-based kernel. As described in the main body, line 8 of Alg 1 is replaced by

$$\varphi(i)[\texttt{prefix\_subwalk}[-1]] + = \left(\prod \texttt{traversed\_edge\_weights}\right) * f(\texttt{length}(\texttt{prefix\_subwalk})), \qquad (16)$$

removing the normalisation factor $p(\texttt{prefix\_subwalk})$.

**Data synthesis**. We consider a synthetic dataset, consisting of a regular $30 \times 30$ mesh graph (900 nodes). We compute a ground truth diffusion kernel $\mathbf{K}_{\text{diff}}^* = \exp(-\beta^* \mathbf{L})$ on this mesh graph with a known length scale $\beta^* = 10$ (hidden from the models), and sample a function from the corresponding GP, shown in Figure 5. Noisy observations are made at 10% of the nodes, indicated by black dots. The task is to predict missing measurements.

**Kernels comparison**. For GP training and inference, we consider three kernels: the exact diffusion kernel $\mathbf{K}_{\text{diff}} = \sigma_f^2 \exp(-\beta \mathbf{L})$, a GRF kernel $\hat{\mathbf{K}}$, and an ad-hoc random walk kernel $\hat{\mathbf{K}}_{\text{ad-hoc}}$ as per Eq. (16). The learned maximum-a-posteriori predictions (posterior mean) are shown in Figure 5 (b)-(d), and the RMSE and NLPD are reported in Table 5. For random walk-based kernels $\hat{\mathbf{K}}$ and $\hat{\mathbf{K}}_{\text{ad-hoc}}$, we sample 10,000 walks per node, truncating any walk exceeding 10 steps. Models are trained using the Adam optimiser with a learning rate of 0.01 for 1,000 iterations.

Clearly, the ad-hoc kernel fails to capture the underlying structure, producing inaccurate predictions. This shows that a principled importance sampling approach is essential for random walk-based kernels to perform well in practice.

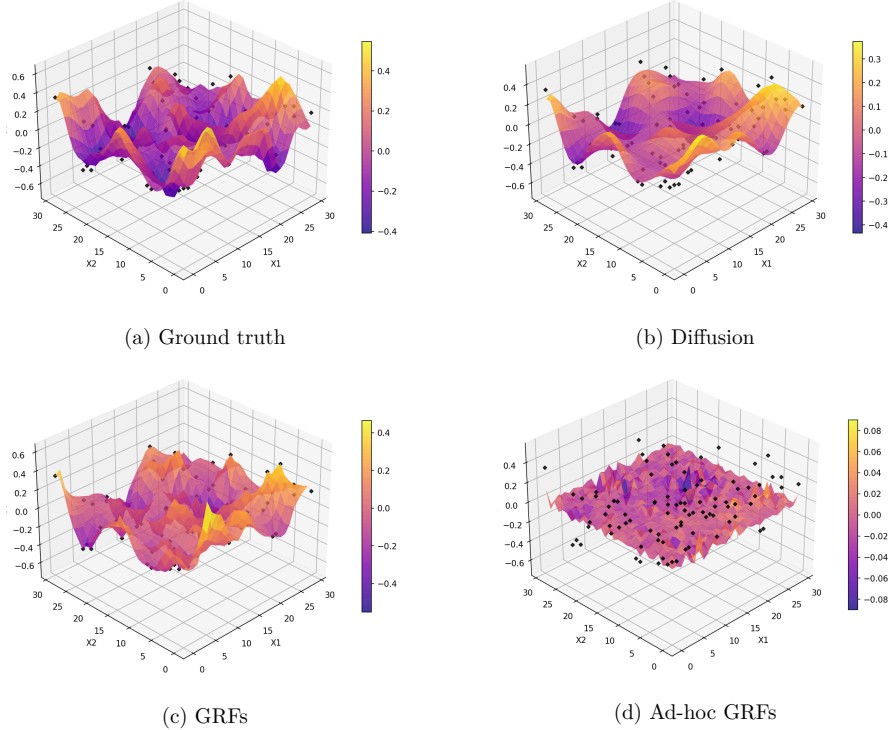

(a) Ground truth

(b) Diffusion

(c) GRFs

(d) Ad-hoc GRFs

Figure 5: **The ad-hoc kernel fails to capture longer-range relationships.** Panel (a): Ground-truth function on a $30 \times 30$ mesh graph; black dots mark noisy observations at 10% of the nodes. Panels (b–d): Posterior means inferred with the exact diffusion kernel, the GRF kernel, and the ad-hoc kernel, respectively. Unlike the principled GRF estimator, the ad-hoc variant produces poor predictions and misses the underlying structure.

Table 5: **The ad-hoc kernel yields much worse predictive accuracy.** Test RMSE and NLPD for the diffusion kernel, principled GRF kernel, and the ad-hoc variant. The ad-hoc kernel exhibits substantially higher RMSE and NLPD.

| Kernel | RMSE | NLPD |
|---|---|---|
| Diffusion | 0.262 | 0.090 |
| GRFs | 0.339 | 0.339 |
| Ad-hoc GRFs | 0.573 | 1.265 |

### C.4 Conjugate gradients convergence

Here, we provide extra discussion to supplement Section 3.1. In particular, we experimentally confirm the $\mathcal{O}(N)$ bound on the condition number of the GRF Gram matrix and the $\mathcal{O}\left(N^{3/2}\right)$ time complexity for solving the corresponding linear system.

Theorem 2 asserts that the condition number of the GRF Gram matrix $\hat{\mathbf{K}}$ is $\mathcal{O}(N)$, assuming that $c := \sum_{r=0}^{\infty} |f_r| \left( \max_{i,j \in [\![1,N]\!]} \mathbf{W}_{ij} d_i/(1-p) \right)^r$ is constant across the class of graphs considered. We empirically confirm that this is the case below in Figure 6. Lemma 1 also uses that the conjugate gradient (CG) method can solve the corresponding linear system in $\sqrt{\kappa\left(\hat{\mathbf{K}} + \sigma_n^2 \mathbf{I}\right)}$ iterations (Shewchuk, 1994), which ultimately unlocks our $\mathcal{O}\left(N^{3/2}\right)$ scaling. This implicitly assumes that we run CG to a fixed error ratio $\varepsilon$ – in our case, $10^{-2}$ – which follows convention for efficient GP methods in the literature (Maddox et al., 2021). We find this CG termination criterion to be empirically robust.

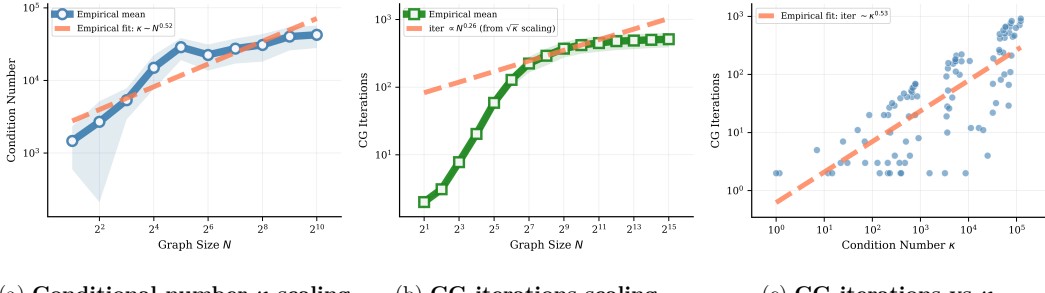

(a) **Conditional number $\kappa$ scaling**  (b) **CG iterations scaling**  (c) **CG iterations vs $\kappa$**

Figure 6: **Empirical scaling behaviour of CG iterations** (a) Condition number vs. graph size for Erdős–Rényi graphs with edge probability $p = \mathcal{O}\left(\frac{1}{N}\right)$. The linear fit in log-log space shows that $\kappa \sim N^{0.52}$, which lies well within the upper bound proved in in Theorem 2, i.e., $\kappa\left(\hat{\mathbf{K}} + \sigma_n^2\mathbf{I}\right)$ is $\mathcal{O}(N)$. (b) CG iterations to solve the $\left(\hat{\mathbf{K}} + \sigma_n^2\mathbf{I}\right)\boldsymbol{x} = \boldsymbol{y}$ to a fixed relative error tolerance $\varepsilon = 10^{-2}$ as N increases. (c) CG iterations plotted directly against $\kappa$. A log-log fit gives iter$\sim \kappa^{0.53}$, which match the expected $\mathcal{O}(\sqrt{\kappa})$ behaviour.

## C.5 REGRESSION TASK: TRAFFIC SPEED PREDICTION

Here we provide further details for the first regression experiment: predicting traffic speeds in the San Jose freeway sensor network (Chen et al., 2001), following the setup of Borovitskiy et al. (2021).

**Dataset**. We use the San Jose freeway sensor network combined with OpenStreetMap data to construct a graph with 1,016 nodes and 1,173 edges (contributors, 2024). Traffic speed measurements (in mph) are available at 325 sensor locations. These values are normalised (zero mean, unit variance), and the data is split into a training set of 250 randomly selected nodes and a test set of the remaining 75 nodes.

**Kernel approximation with GRFs**. We used two variants of GRFs kernels. The first GRF kernel uses a diffusion-shape modulation function $f_l = \frac{(-\beta/2)^l}{l!}$. This is a truncated power series expansion of the diffusion kernel, where the learnable hyperparameters are length scale $\beta$ and kernel variance $\sigma_k^2$. The second kernel directly learns the modulation coefficients $(f_l)_{l=0}^{\infty}$, which are initialised randomly and learned via log marginal likelihood. For both GRF variants, we fix $p_{\text{halt}} = 0.1$ and truncate walks at a maximum length of 10, and vary the number of walks per node $n \in \{1, 2, 4, ..., 8192\}$. Since the traffic network contains roughly 1,000 nodes, we also include the exact diffusion kernel $\mathbf{K}_{\text{diff}}$ as a baseline. The kernel configurations are:

$$\textbf{Exact Diffusion}: \quad \mathbf{K}_{\text{diff}} = \sigma_f^2 \exp(-\beta\mathbf{L}),$$

$$\textbf{Diffusion-shape } \hat{\mathbf{K}}: \quad f_l = \frac{(-\beta/2)^l}{l!},$$

$$\textbf{Fully-learnable } \hat{\mathbf{K}}: \quad f_l \text{ learned directly.}$$

**Regression task**. We apply GP inference using the 250 labeled nodes as training data to predict traffic speeds at all 1,016 nodes in the network. The kernel hyperparameter and noise variance $\sigma_n^2$ are learned by maximising the log marginal likelihood, using Adam. Posterior inference then yields the predicted mean $\hat{\boldsymbol{\mu}}$ and covariance $\hat{\boldsymbol{\Sigma}}$ of the latent traffic speed function over the graph.

To quantify accuracy, we compute the negative log probability density (NLPD) and root mean squared error (RMSE) on the 75 test nodes between the true speeds $\boldsymbol{y}^{\text{test}}$ and the MAP estimate $\hat{\boldsymbol{\mu}}$:

$$\text{RMSE} = \sqrt{\left(\frac{1}{N_{\text{test}}}\right) \sum_{i=1}^{N_{\text{test}}} (\hat{\mu}_i - y_i)^2}$$

$$\text{NLPD} = -\left(\frac{1}{N_{\text{test}}}\right) \sum_{i=1}^{N_{\text{test}}} \log p(y_i \mid x_i, D_{\text{train}})$$

The experiment is repeated five times with different random seeds. The results are shown in Figure 3 (a)-(b) in the main text.

**Capturing global and local patterns**. Using the visualisation toolkits by Borovitskiy et al. (2021), we illustrate the GRF-GPs posterior inference results on the San Jose traffic network in Figure 7. The left panel provides a global view over the full network, while the right panel zooms in on a specific highway junction. We observe that the global inferred mean (top left) captures large-scale spatial variation across the network—speeds are higher on main freeway segments and lower in peripheral or downtown regions. Notably, in the zoomed-in view (top right), the model successfully distinguishes speeds across tightly packed lanes running in opposite directions. Despite spatial proximity, the posterior assigns significantly different mean values to adjacent but directionally distinct segments, demonstrating that GRF-GPs capture connectivity-aware patterns rather than relying solely on Euclidean distance. The bottom row visualises posterior uncertainty, with standard deviation plotted over the full graph (bottom left) and zoomed in section (bottom right). These results confirm that GRF-GPs respect both global graph structure and local topology, delivering interpretable and spatially coherent predictions on complex, real-world networks.

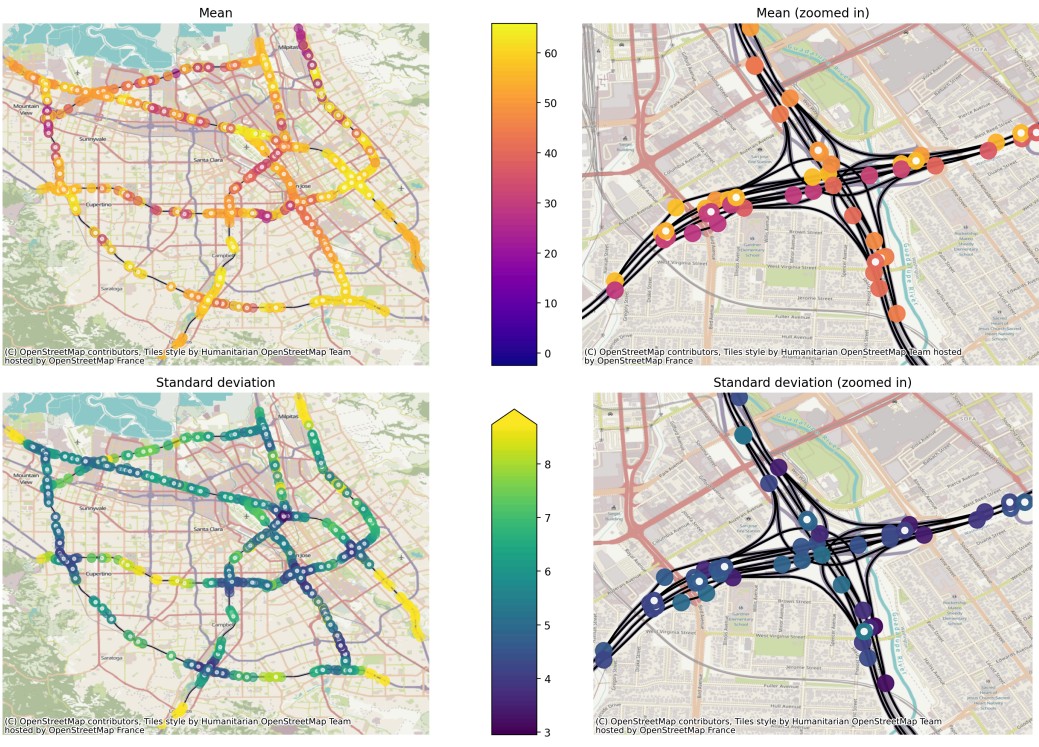

Figure 7: Posterior inference using GRF-GPs on the San Jose traffic network. **Top left**: Mean predictions across the full graph. **Top right**: Zoomed-in directional differences between closely spaced lanes. **Bottom left**: Posterior uncertainty over the network. **Bottom right**: Zoomed view reveals local variation in confidence. Coloured dots are sensor nodes; white dots indicate training nodes.

C.6 REGRESSION TASK: WIND VELOCITY INTERPOLATION

Here we provide further details about the wind velocity interpolation task from the ERA5 dataset (Hersbach et al., 2019). Our problem setup follows that of Wyrwal et al. (2024) and Robert-Nicoud et al. (2023).

**Dataset**. We use the average wind velocity field from the ERA5 dataset at three altitudes: 0.1 km, 2 km, and 5 km. The surface of the globe (formally, the manifold $S^2$) is discretised at a resolution of 2.5° longitude by 2.5° latitude, yielding a $k$-nearest neighbours graph $\mathcal{G}$ with roughly 10K nodes, on which we apply our scalable GRF-GPs algorithm. The task is to predict the velocity fields on the held-out test nodes. The locations along the Aeolus satellite track (1441 nodes) serve as training data, while all remaining nodes are treated as the test set.

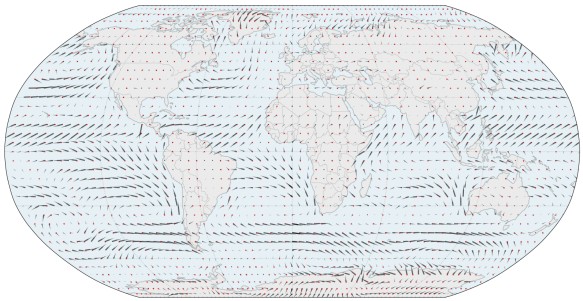

Figure 8: Ground-truth wind velocity field from the ERA5 dataset at 0.1 km above sea level. Black vectors show local wind velocities. Red dots mark 1441 Aeolus satellite track locations, used as training data in the interpolation task (Reitebuch, 2012).

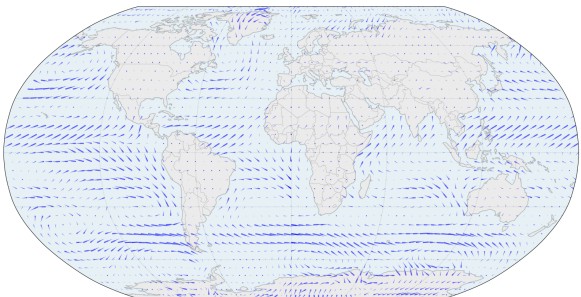

Figure 9: Predicted wind velocity field using GRF-GPs. Blue vectors represent MAP predictions (GP posterior mean).

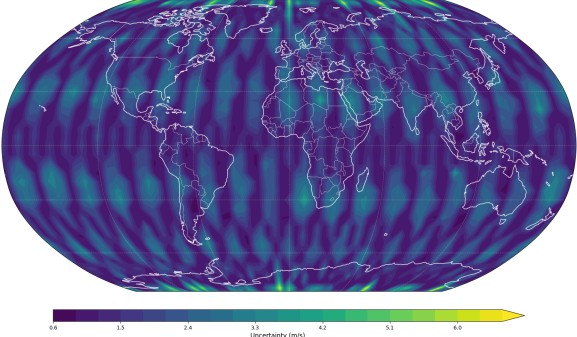

Figure 10: Prediction uncertainty (GP posterior covariance) using GRF-GPs. Brighter regions indicate higher uncertainty, which is significantly reduced near satellite track.

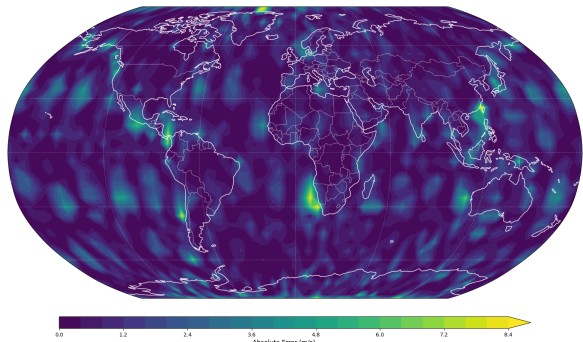

Figure 11: Absolute error between ground-truth and MAP-predicted velocities. GRF-GPs achieve accurate predictions, with error patterns aligned with uncertainty estimates.

**Experiment setup**. We use the fully-learnable and diffusion-shape GRF kernels, varying the random walk budget, similar to the method described in App. C.5. Note the exact diffusion kernel $\mathbf{K}_{\text{diff}}$ cannot be applied on this large graph. We measure the NLPD and RMSE to evaluate the kernel performance. The results are shown in Figure 3 (c-d) in the main text.

**Uncertainty-aware wind velocity interpolation**. Figures 8–11 visualise GRF-GPs inference on the ERA5 wind dataset at 0.1 km altitude. For visualisation clarity, the $k$-nearest neighbour graph on the globe is downsampled. Figure 8 shows the ground-truth wind field with training node positions marked in red. Figure 9 shows the MAP prediction, and Figure 10 shows the posterior uncertainty, which is notably reduced along the Aeolus satellite track. Finally, Figure 11 displays the absolute error field.

### C.7 LARGE SCALE BAYESIAN OPTIMISATION ON GRAPHS

Here, we describe our evaluations of the performance and scalability of GRF-GPs on Bayesian optimisation (BO) tasks, as detailed in Section 4.3. We test the methodology across three settings: (1) four synthetic graph benchmarks, (2) four real-world social network datasets for identifying influential users, and (3) three wind interpolation datasets. First, let us describe the benchmark datasets.

**1. Synthetic benchmarks**. We consider four synthetic graph benchmarks.

- **Unimodal function on grid**: a function with a smooth central peak, discretised on a $1000 \times 1000$ grid graph.
- **Multi-modal function on grid**: a function with several randomly placed peaks, discretised on a $1000 \times 1000$ grid graph.
- **Community graph**: a community graph generated via a stochastic block model (SBM), with nodes in a community $C_i$ assigned a score by sampling from $\mathcal{N}(\mu_i, \sigma_i^2)$.
- **Circular graph**: a sinusoidal function defined on a ring, discretised into a $k$-nearest neighbour graph with $10^6$ nodes.

All signals are perturbed with Gaussian noise ($\sigma_n^2 = 0.1$). Random features $\mathbf{\Phi}$ are computed with 100 walks per node, with halting probability $p_{\text{halt}} = 0.1$. Random walks longer than 5 hops are truncated.

**2. Social networks benchmarks: identify the most influential user**.

We consider four real-world social network datasets (Table 6) from the Stanford Network Analysis Project (SNAP) (Leskovec and Krevl, 2014), with up to 1.1M nodes. Each node represents a user in the network. Following Wan et al. (2023), we use node degree as a proxy for user influence, and the task is to identify the most 'influential' users in each network.

Table 6: Summary of four SNAP datasets used for large-scale BO experiments. Each dataset corresponds to a user-level social network, with node degree used as a proxy for influence.

| Dataset | Nodes | Edges | Maximum Degree | Description |
|---------|-------|-------|----------------|-------------|
| YouTube | 1,134,890 | 2,987,624 | 28754 | Youtube online social network |
| Facebook | 22,470 | 171,002 | 709 | Facebook page-page network with page names. |
| Twitch | 168,114 | 6,797,557 | 35279 | Social network of Twitch users. |
| Enron | 36,652 | 183,831 | 1383 | Email communication network from Enron |

**3. ERA5 wind velocity field: predict the location with greatest wind speed**.

To demonstrate the utility of GRFs for BO on manifolds, we use the ERA5 wind datasets at three altitudes. Full details of dataset processing are provided in App C.6.

**Algorithm Baselines**. We compare GRF-based Thompson sampling against three search heuristics:

- **Random search**: uniformly samples nodes without replacement.
- **Breadth-first search (BFS)**: sequentially expand observed nodes along the adjacency structure in breadth-first order.
- **Depth-first search (DFS)**: sequentially expand observed nodes along the adjacency structure in depth-first order.

**BO setting**. In each experiment, algorithms are initialised with up to 1,000 samples and then run for up to 1,000 BO iterations, repeated across five random seeds. At each iteration, we report *simple regret*, defined as the difference between the global maximum and the best function value observed so far.

---

**Algorithm 3**: Graph Thompson Sampling with GRFs

---

1   **Inputs**: black-box function $h$, candidate nodes `x_all`, initial sample size `N_0`, number of BO steps `T`.

2   **Output**: augmented dataset (`x_obs`, `y_obs`).

3   initialise `x_obs` $\leftarrow \{x_i\}_{i=1}^{\text{N\_0}}; x_i \sim \text{Unif}(\text{x\_all})$

4   initialise `y_obs` $\leftarrow \{h(x_i) + \varepsilon_i\}_{i=1}^{\text{N\_0}}$

5   **for** $t = 1, ..., \text{T}$

6     `model.train(x_obs, y_obs)`

7     `s_t` $\leftarrow \text{PosteriorSample}(\text{model}, \text{x\_all})$

8     `x_t` $\leftarrow \text{ArgMax}(\text{s\_t})$

9     `y_t` $\leftarrow h(\text{x\_t}) + \varepsilon$

10    `x_obs` $\leftarrow$ `x_obs` $\cup$ `x_t`

11    `y_obs` $\leftarrow$ `y_obs` $\cup$ `y_t`

12   **end for**

13   **return** (`x_obs`,`y_obs`)

---

C.8   Classification task: Cora citation network

Here we provide more experimental details about the classification task on the Cora scientific citation network (McCallum et al., 2000). This experiment highlights the application of GRF-GPs in a more challenging, non-conjugate inference setting.

**Dataset and preprocessing**. The Cora dataset is a standard benchmark in graph-based machine learning. It consists of a citation network, where each node corresponds to a scientific publication and each edge represents a citation. Each publication is labelled with one of seven machine learning topics (Figure 12). While Cora also includes textual features, we focus solely on the graph structure. We extract the largest connected component of the citation graph, resulting in a subgraph with 2,485 nodes and 5,069 edges.

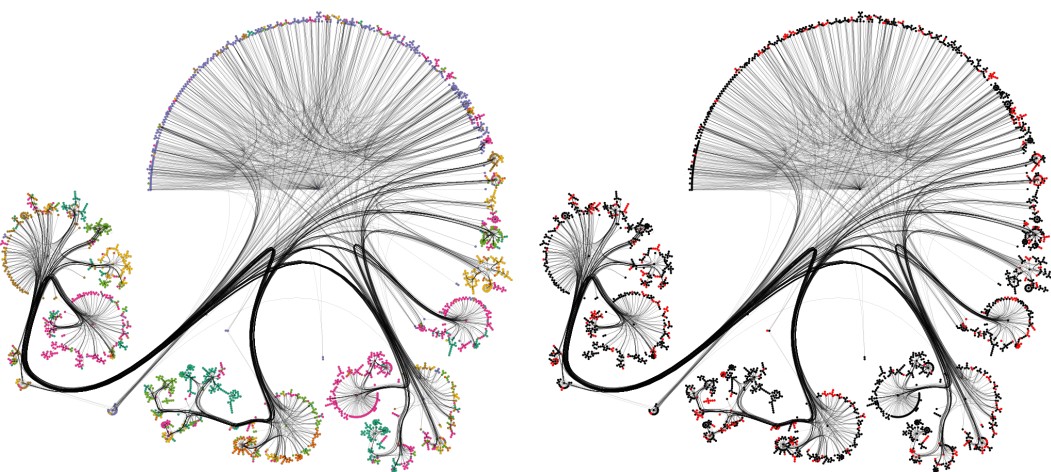

(a) Ground-truth labels. Each color denotes a class

(b) Prediction errors using graph GP with Matérn kernel. Red nodes are misclassified

Figure 12: Cora dataset classification with graph GP.

**Sparse variational inference for classification**. In classification tasks, the likelihood functions are usually non-Gaussian (softmax), so the posterior is not analytically tractable. Denote the $N_{\text{train}}$ training nodes as $\boldsymbol{x}$ and the $M$ inducing nodes as $\boldsymbol{z}$. Define latent function values at the training inputs as $\boldsymbol{h} = (h(x) : x \in \boldsymbol{x})$ and function values at inducing nodes as $\boldsymbol{u} = (u(z) : z \in \boldsymbol{z})$. Assume a GP prior $p(\boldsymbol{u}) = N(\boldsymbol{0}, \mathbf{K}_{uu})$ and a likelihood $p(y_i \mid h_i)$ (softmax). Choose a Gaussian variational posterior $q(\boldsymbol{u}) = N(\boldsymbol{\mu}, \boldsymbol{\Sigma})$ and induce the marginal $q(\boldsymbol{h}) = \int p(\boldsymbol{h} \mid \boldsymbol{u}) q(\boldsymbol{u}) d\boldsymbol{u}$. Under this approximation we maximise the evidence lower bound (ELBO): $\mathcal{L}_{\text{ELBO}} = \sum_{i=1}^{N_{\text{train}}} E_{q(h_i)}[\log p(y_i \mid h_i)] - \text{KL}(q(\boldsymbol{u}) \parallel p(\boldsymbol{u}))$. This variational treatment replaces the intractable posterior with a tractable family and supplies a principled objective (a lower bound on $\log p(\boldsymbol{y})$); it yields coherent predictive distributions by integrating over $q(\boldsymbol{h})$ rather than relying on point approximations, which is especially important when the likelihood breaks conjugacy.

**Experiment setup**. We compare classification accuracy across exact kernels (diffusion and Matérn) and the GRF kernel. We use an 80/20 train-test split on the largest connected component of the graph. The goal is to predict the class labels of all nodes based on the graph structure alone. All models are trained using softmax likelihood. Optimisation is performed for up to 1000 iterations using the Adam optimiser. To reduce uncertainty and assess variability, each configuration is repeated five times with different random seeds. We also measure the sparsity of the resulting GRF kernels. Results are reported in Table 7, showing that with a sufficient number of random walkers, the flexibility of the GRF kernel allows it to capture the graph structure effectively and outperform the exact kernel baselines.

## C.9 WHY AREN'T SVGPS EFFICIENT ON GRAPHS?

In this section, we briefly compare our method to *sparse variational Gaussian processes* (SVGPs) (Titsias, 2009), a different efficient GP framework popular for problems in Euclidean space. However, a core difficulty with applying SVGPs to graphs is the fact that, in order to compute a smaller kernel matrix at some subset of *inducing points*, we still need to compute the entire Gram matrix. This is in general $\mathcal{O}(N^3)$.

Table 7: **The GRF kernel reaches highest accuracy in the Cora benchmark**. Classification accuracy on the Cora dataset with different graph kernels. With $n = 16384$ walks per node (22.17% non-zero entries), the GRF kernel outperforms the diffusion and Matérn kernels (Borovitskiy et al., 2021).

| Kernel | Form | Accuracy |
|--------|------|----------|
| Diffusion | $\mathbf{K}_{\text{diff}} = \exp(-\beta\mathbf{L})$ | $85.31 \pm 0.61\%$ |
| GRFs | $\hat{\mathbf{K}} = \mathbf{\Phi}\mathbf{\Phi}^\top$ | $\mathbf{87.04 \pm 0.53\%}$ |
| Matérn | $\mathbf{K}_{\text{Matérn}} = \left(\frac{2\nu}{\kappa^2} + \tilde{\mathbf{L}}\right)^{-\nu}$ | $86.72 \pm 0.31\%$ |

**Comparison with Euclidean setting**. Suppose $\mathcal{V}$ denotes the full set of $N$ training datapoints, and $\mathcal{V}_{\text{ind}} \subset \mathcal{V}$ denotes a subset of $m$ inducing points. For e.g. the Gaussian kernel, one can compute

$$\mathbf{K}_{\text{ind.}} = \left[\exp\left(-\frac{(\boldsymbol{x}_i - \boldsymbol{x}_j)^2}{2\sigma^2}\right)\right]_{i,j \in \mathcal{V}_{\text{ind}}} \in \mathbb{R}^{m \times m} \tag{17}$$

in $\mathcal{O}(m^2)$ time and space complexity. One need not compute the kernel at points in $\mathcal{V} \setminus \mathcal{V}_{\text{ind}}$. Conversely, for graphs, it is not straightforward to compute some specific $[\mathbf{K}_\alpha]_{ij}$ without first materialising the full $N \times N$ matrix

$$\mathbf{K}_\alpha(\mathbf{W}) = \sum_{r=0}^\infty \alpha_r \mathbf{W}^r, \quad \alpha_r \in \mathbb{R} \ \forall r \in \{0, 1, ..., \infty\} \tag{18}$$

where $\mathbf{W}$ is the adjacency matrix for the *entire* graph $\mathcal{G}$. One can then extract the corresponding subset of entries corresponding to the $m$ inducing points. This means that methods like SVGP may not actually provide time complexity gains, since we are still bottlenecked by $\mathcal{O}(N^3)$ to compute the kernel at the $m$ inducing points in the first place.

**Extra baseline**. Notwithstanding the above, we can implement SVGPs on graphs to compare performance to GRFs, even if this baseline may not be truly efficient in practice. On the traffic speed prediction task, we trained SVGP models using graph diffusion kernel with 150 inducing points. We used Adam optimiser with a learning rate of 0.01 for 1000 steps. The results are shown in Figure 13. As expected, SVGP underperforms compared to the exact kernel. GRFs perform better even with a modest number of walkers.

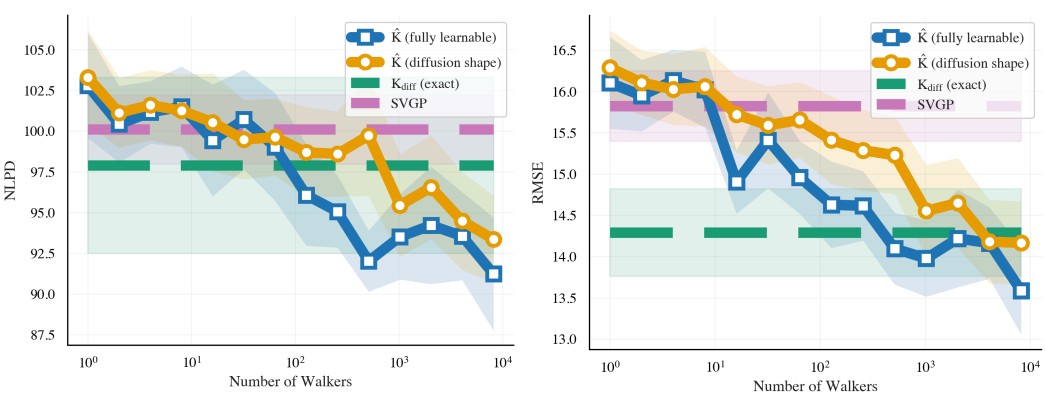

(a) Traffic NLPD with SVGP baseline

(b) Traffic RMSE with SVGP baseline

Figure 13: **Extra SVGP baseline**. Companion results to Figure 3, with additional SVGP baseline. SVGP as expected performs worse than the $\mathbf{K}_{\text{diff}}$ baseline in both NLPD and RMSE. GRFs achieve better NLPD & RMSE beyond $\sim 100$ walkers.

