# OpenReview forum: "Graph Random Features for Scalable Gaussian Processes"
_ICLR.cc/2026/Conference — ICLR 2026 Poster_

### Official Review · Reviewer_e8Qw · 2025-10-30

**Soundness:** 3
**Presentation:** 4
**Contribution:** 2
**Rating:** 6
**Confidence:** 5

**Summary:**

This paper applies the random feature approach in kernel-based learning to graph problems (such as node classification/regression). For context, graph random features are a recently introduced stochastic estimator of a family of kernel matrices defined on graphs. The authors use these features for node regression and Bayesian optimization. They show an $O(N^{3/2})$ time complexity for kernel parameter estimation and demonstrate that a learnable kernel with random features outperforms the graph diffusion kernel, either in the full matrix form or in its random feature representation.

**Strengths:**

A main bottleneck of Gaussian processes is the $O(N^3)$ scalability barrier. The proposed approach provably reduces the cost to $O(N^{3/2})$. Additionally, the authors demonstrate experiments to show that they can handle 1M graph nodes comfortably by using a single GPU.

The writing of the paper is excellent and easy to follow, balancing clarity and conciseness for graph random features (GRFs), which contain many subtle details.

(Half strength, half weakness) GRFs are an interesting extension of random features from the Euclidean domain to the graph domain. It is pleasant to see the use of GRFs in machine learning tasks. On the other hand, this paper mostly concerns applying the concept, weakening the contribution. Moreover, as graph neural networks (GNNs) become so widely used for the tasks concerned in this paper, the advantages of GRFs are unclear.

**Weaknesses:**

On the theoretical side, given the prior work on GRFs and the machinery to perform GP inference and parameter estimation by using random features, the contribution of this work primarily lies in applying the former to the latter. A majority of the background for this application is known (e.g., the concentration of GRF in Theorem 1, the parameter estimation method in Eqn (8)--(11), and the posterior inference in Eqn (12)). To be fair, the authors do provide new results, such as the $O(N^{3/2})$ complexity for inference due to the $O(N)$ spectral norm of the kernel, but this contribution is less significant than the establishment of the GRF itself.

On the practical side, it is unclear how kernel/GP methods fare with GNNs. There are no such experiments in the paper. GNNs scale better than kernels and they are data-efficient for node-level problems. For example, for the Cora graph and the classification task that the authors discuss in Section 4.4, GNN can achieve over 80% accuracy with 20 training nodes per class (a labeling rate of 5.2% only). Can GRF be as data-efficient?

Another practical question is the Bayesian optimization over graphs, which has a finite search space. Since predicting the graph signal on each node is cheap and relatively accurate by using a GNN given a small training set, a natural heuristic is that one does not use Thompson sampling for the acquisition but only picks the best GNN predictions. The practical value of the paper can be strengthened by experimenting with such a simple baseline.

**Questions:**

See comments above.

---

> ### Author Response · Authors · 2025-11-18
> **Rebuttal**
>
> We thank the reviewer for their comments on the text. We are pleased they find the writing clear. We address all concerns and questions below.
>
> 1. **Core contributions**. The reviewer is correct to note that our core contribution is applying GRFs to the novel application of graph-based GPs and BO -- a contribution we believe to be interesting and impactful. We are pleased that they recognise our new complexity results.
>
> 2. **Comparison to GNNs.** GNNs are indeed strong baselines for scalable prediction on graphs, but they serve a different purpose from our GRF–GP framework, which is designed for *probabilistic* modelling with calibrated uncertainty. Nonetheless, we agree that including a GNN baseline may be interesting, so have added this as an **extra baseline**. To provide a fair comparison, we trained a 3-layer Graph Convolutional Network (GCN) with hidden dimensions = [64, 32, 16] on the traffic regression task. As shown in Figure 13 in the Appendix D.3, the GCN performs slightly worse than GRFs in terms of RMSE.
>
> 3. **GNNs for BO**. Thanks for the comment. We think the reviewer is suggesting to train a GNN on a subset of data, make predictions for missing nodes, make an observation at the node with the greatest prediction, retrain, and repeat. Is this the case? Using a GNN as a deterministic surrogate with greedy sampling seems likely to exploit rather than explore; we suggest that uncertainty-aware BO strategies are likely to perform better. However, if the reviewer could kindly clarify whether this is what they have in mind and if possible cite some related papers, we would be happy to try implementing this idea as an extra baseline!
>
> 4. **Data efficiency on Cora**.  Thank you for the question. Graph GP models can indeed be very data-efficient. For example,Borovitskiy et al. report that a Matérn Gaussian Process on graphs achieves 79% accuracy on the Cora citation network using only 140 labelled nodes (avg 20 per class), demonstrating strong performance even in low-label regimes [1]. This highlights that GP-based approaches can complement GNNs when label efficiency and calibrated uncertainty are important.
>
> We again thank the reviewer, and warmly invite them to respond. In light of our clarifications and the **extra GCN baseline**, we hope they might consider raising their score.
>
> [1] Borovitskiy et al., *Matérn Gaussian Processes on Graphs*, AISTATS 2021.

---

### Official Review · Reviewer_RsGB · 2025-10-31

**Soundness:** 3
**Presentation:** 3
**Contribution:** 3
**Rating:** 6
**Confidence:** 3

**Summary:**

This paper proposes Graph Random Features (GRFs) to construct learnable graph node kernels which estimate the sparse GP covariance function reducing the cubic computational complexity of exact GPs and applying these graph-based approximation technique for sparse GP estimation for Bayesian optimization via Thompson sampling. The contributions of this paper are considerably novel, as they build upon the previous work of Reid et. al. on GRFs specifically into GP domain and then provide an application of this technique by applying it into a BO setting showcasing the effectiveness and practicality.

**Strengths:**

The paper presents a well-motivated contribution that bridges recent advances in graph kernels and scalable GP inference. The use of GRFs for graph node kernels is an innovative idea that leverages random walks to approximate covariance structures efficiently. The theoretical development is sound and neatly connects GRFs with probabilistic kernel approximation, culminating in a clear argument for the claimed time complexity. The results claimed in Theorem2 and Lemma1 also establish theoretical underpinnings of the proposed work, contributing to the theoretical understanding of GRFs.

Empirically, the paper demonstrates wall-clock speedups and scalability to graphs with >1M nodes, which is impressive for a sparse GP-based method. The reported experiments, on regression and BO tasks, are diverse enough to illustrate both predictive and computational benefits.

Moreover, the paper is clearly written and well-organized. Overall, the paper is theoretically and empirically solid and easy to follow for community working on GPs and BO.

**Weaknesses:**

While the core idea of the paper is novel, the experimental validation somewhat lacks depth and breadth. The comparisons are limited mainly to diffusion kernels and do not include strong baselines such as sparse variational GPs, approximation based on inducing-point (support set). Without these benchmarks, it is difficult to judge the real-world competitiveness of GRFs in GP regression and BO tasks. The paper also does not sufficiently explore the sensitivity of the method to hyperparameters such as the number of random walks, halting probability, or walk length.

Moreover, the paper does not provide bounds on the GP posterior approximation error, so while kernel estimation is shown to be accurate, the implications for predictive uncertainty are less clear. Empirical metrics focus on RMSE and NLP, but uncertainty calibration and robustness are not analyzed. The BO results show consistent performance gains, but the lack of statistical testing limits the strength of these claims to some extent.

**Questions:**

I have few concerns.
-- The assumption that the number of random walkers 𝑛 is independent of 𝑁 is theoretically elegant but practically fragile. In dense or long-range graphs, a small 𝑛 can lead to high-variance kernel estimates and poor posterior calibration.
-- The sparse approximation $\hat{K} = \Phi \Phi^{\top}$ is unbiased only in expectation, but in practice, the variance of the estimator is not analyzed in sufficient depth. The paper cites exponential concentration (Theorem~1) but never empirically validates how variance scales with $n$ or the graph topology. As a result, while computational speed is improved, the fidelity of posterior uncertainty (a key factor in Bayesian inference) may degrade unpredictably.

-- The footnote in Section~2 admits an $\mathcal{O}(1/n)$ bias for diagonal entries and notes  that removing it breaks positive definiteness. Yet, the authors use the biased estimator throughout without quantifying the impact of this bias
on GP likelihood or predictive uncertainty. This is a non-trivial issue --- GP models are sensitive to kernel conditioning,
and even small biases can affect the posterior mean and variance.

---

> ### Author Response · Authors · 2025-11-18
> **Rebuttal**
>
> We thank the reviewer for their comments, and are happy they find the paper well-motivated and the empirical results impressive. We are pleased they find the experiments diverse. We address all their concerns and questions below.
>
> 1. **Baselines**. The usual strong GP baselines cannot be straightforwardly applied to graphs. The core issue is that, in order to compute the kernel between a subset of $m$ points, one still needs to compute the _entire_ exact kernel $K = \sum \alpha_r W^r$ before extracting the relevant rows and columns. This still in general requires $O(N^3)$ time. This differs from the Euclidean setting which is $O(m^2)$, since here you can just compute e.g. $\exp(-(x_i - x_j)^2)$ for every pair of inducing points. As such, efficient methods that work well in Euclidean space may not actually provide speed benefits in the discrete setting. However, despite the bottleneck, we can still in principle compare GRFs to SVGP on sufficiently small graphs as an interesting extra baseline. **We have now added this extra experiment to App. D.3.** We compare GRFs against an SVGP model equipped with the exact diffusion kernel on the traffic dataset. SVGP underperforms GRFs on both RMSE and NLPD.
>
> 2. **Hyperparameter sensitivity.** We provide ablations over the key hyperaparameter, the number of walkers, in Figs 2 and 3. The maximum walk length is mainly an implementation detail; it can be chosen to be sufficiently long that all walkers terminate beforehand with high probability. Previous papers have studied the impact of the termination probability on the kernel estimator [1]; naturally, a lower probability reduces the estimator variance, typically with diminishing returns for very long walks (e.g. provided the graph spectral radius is not too large).
>
> 4. **Bounds on the posterior approximation error.** Thanks for the great question. On account of the inverse, relating the pointwise concentration of $K$ to the quality of the posterior approximation is a difficult, open research question. Avron (2017) [2] shows how the quality of approximation of $K$ in the _spectral_ sense (see Eq. 2 of this paper) is related to accuracy guarantees of random feature based kernel methods on downstream tasks such as kernel ridge regression and kernel k-means clustering. The quality of the spectral approximation can in turn be related to the kernel estimator MSE [3]. But a full theoretical analysis of how pointwise estimation relates to e.g. the KL divergence to the true posterior remains (to our knowledge) unknown, and would be a great topic for a future theoretical paper!
>
> 4. **Variance of the kernel estimate**. Please see the papers _Quasi-Monte Carlo Graph Random Features_ [4] and _Variance-Reducing Couplings for Random Features_ [5] for detailed studies of the variance of the GRF kernel estimate (including techniques to reduce it). Whilst interesting, these involved mathematical questions are not the chief focus of our methodology-driven work. In practice, our empirical results show that a modest number of walkers gives low enough kernel estimator variance for strong experimental performance; there is a tradeoff between cost and performance, made clear by e.g. Fig 3. We also remark that any given draw of random features implicitly defines a kernel, so even if the variance of the approximation of the corresponding (learned) function of the adjacency matrix is high we often achieve good experimental performance in practice.
>
> We again thank the reviewer. We hope that, in light of these clarifications and the extra SVGP baseline, they might kindly consider raising their score. We invite them to respond with any further questions.
>
> [1] Taming graph kernels with random features, Krzystof Choromanski, ICML 2023
> [2] Random Fourier Features for Kernel Ridge Regression: Approximation Bounds and Statistical Guarantees, Avron et al., ICML 2017
> [3] The Geometry of Random Features, Choromanski et al., AISTATS 2018
> [4] Quasi-Monte Carlo Graph Random Features, Reid et al., NeurIPS 2023
> [5] Variance-Reducing Couplings for Random Features, Reid et al., ICLR 2025

---

### Official Review · Reviewer_sDRg · 2025-10-31

**Soundness:** 3
**Presentation:** 4
**Contribution:** 3
**Rating:** 6
**Confidence:** 3

**Summary:**

Traditional GPs on graphs suffer from cubic time complexity (O(N^3)), limiting their scalability to small graphs. This paper leverages graph random features, which approximate graph kernels via sparse random walk–based feature maps, enabling sub-quadratic inference with strong theoretical guarantees. GRFs yield unbiased kernel estimates with exponential concentration bounds, allowing the GP covariance matrix to be approximated efficiently while preserving accuracy.

**Strengths:**

1. The authors provide rigorous complexity analysis and concentration bounds, proving that GRF-based GPs achieve ($O(N^{3/2})$) time complexity with probabilistic accuracy guarantees — a solid theoretical contribution that enhances credibility.
2. The experiments are diverse, covering both synthetic and real-world tasks. Results consistently support the theoretical claims and demonstrate substantial speedups without significant performance loss.
3. Despite the technical depth, the paper is well-structured and clearly written. Figures, pseudocode, and explanations make the ideas accessible.

**Weaknesses:**

1. The empirical evaluation focuses mainly on regression and Bayesian optimization; classification and non-conjugate inference are mentioned but not fully explored. This somewhat limits the generality of the method’s demonstrated usefulness.
2. The performance of GRFs depends on parameters like the number of random walks and the maximum walk length. Although heuristics are provided, the sensitivity to these settings could be further studied or automated.
3. While the paper compares dense and sparse GRFs, it does not extensively benchmark against other scalable GP frameworks (e.g., inducing point methods, variational GPs, or Kronecker-structured GPs), which would strengthen claims of superiority.

**Questions:**

1. Clarification on Practical Hyperparameter Sensitivity

The performance of GRFs depends on parameters such as the number of random walks, the halting probability, and the maximum walk length. Appendix C.1 partially answers Question 1, but is limited to empirical and qualitative descriptions. Could the authors provide more quantitative guidance on how these hyperparameters influence the trade-off between accuracy, sparsity, and computational cost? For example, is there an empirical scaling rule or automatic tuning strategy that generalizes across datasets?

2. Comparison with Other Scalable GP Frameworks

While the paper contrasts dense vs. sparse GRFs, it does not directly compare with other established scalable GP techniques (e.g., SVGP, SKI, or Kronecker GPs). Could the authors comment on how GRFs compare empirically or theoretically to these alternatives, especially in terms of scalability and approximation accuracy?

3. Clarification on Bias and Positive-Definiteness

Section 2 mentions a small $O(1/n)$ bias in diagonal kernel entries due to shared randomness among random walks. Could the authors clarify whether this bias ever affects posterior stability or positive definiteness in practice? Have they observed numerical issues during GP training? A short empirical note or ablation (e.g., comparing one vs. two independent random walk ensembles) would help readers understand if this bias is purely theoretical or occasionally relevant.

---

> ### Author Response · Authors · 2025-11-18
> **Rebuttal**
>
> We thank the reviewer for their detailed reading of the text. We are pleased they find the theory rigorous and the experiments diverse, and are happy to address all questions and concerns below.
>
> 1. **Classification and non-conjugate inference**. This initial paper mainly focuses on regression and BO since pathwise conditioning is still an open research direction for classification. We agree that this is an interesting direction for future work.
>
> 2. **Hyperparameters**. Section C.1 provides guidance for choosing GRF hyperparameters like the number of walkers. Section C.3 provides detailed ablations to isolate the effects of different aspects of GRFs. The maximum walk length is mainly an implementation consideration; in practice, one can make it sufficiently large that all walkers terminate beforehand with high probability. More quantitatively, given $n$ terminating walkers, with probability at least $1-\delta$, the GRF for node $i$ will have $n \log(1 - (1-\delta)^{1/n}) \log(1-p)^{-1}$ or fewer nonzero entries, which upper bounds the cost of matrix-vector multiplication at each CG iteration. Meanwhile, the condition number is bounded by a constant multiplied by $N$, as described in the main text. This relates the number of walkers to the GRF sparsity and thus the computational cost. We will make sure this is clear in the paper. Thanks. We agree that automatic tuning strategies provide an exciting direction for future work.
>
> 2. **Comparison with other scalable GP frameworks**. Scalable GP frameworks are less well-understood for graph structured data. For example, in order to compute the kernel between a subset of $m$ points, one still needs to compute the _entire_ exact kernel $K = \sum \alpha_r W^r$ before extracting the relevant rows and columns. This still in general requires $O(N^3)$ time, unlike in the Euclidean setting which is $O(m^2)$ (since here you can just compute e.g. $\exp(-(x_i - x_j)^2)$ for every pair). Despite the $O(N^3)$ bottleneck, we can still compare GRFs to SVGP as an extra (non-scalable) benchmark. We have added a **new experiment** comparing GRFs against an SVGP model equipped with the exact diffusion kernel on the traffic dataset. SVGP underperforms GRFs on both RMSE and NLPD. We provided the results in Appendix D.3.
>
> 3. **Clarification on bias and positive definiteness**. In fact, the shared randomness is needed to _guarantee_ positive definiteness. One could instead use two different ensembles for unbiased estimation, at the cost of _losing_ this PSD guarantee. Shared walks work much better in practice and give more stable posterior estimates, as also reported in other settings in the literature [1]. Thanks for the interesting question.
>
> We thank the reviewer again for their thoughtful feedback. In light of our updates, we hope they might consider raising their score. We warmly invite them to respond with any further questions.
>
> [1] Variance-Reducing Couplings for Random Features, Reid et al., ICLR 2025

---

### Official Review · Reviewer_MPVY · 2025-11-02

**Soundness:** 3
**Presentation:** 4
**Contribution:** 3
**Rating:** 8
**Confidence:** 4

**Summary:**

The authors propose a scalable GP methodology for discrete input spaces using graph random features (GRFs) in combination with conjugate gradient (CG) methods and Hutchinson’s trace estimator. A theoretical analysis is presented showing that the computational complexity of the proposed approach scales as $O(N^{3/2})$, which is a significant improvement over the $O(N^3)$ complexity when using exact kernels. Extensive numerical studies are presented to illustrate the performance of the proposed approach.

**Strengths:**

The paper is well-written and organized. Relevant background and motivation is discussed clearly and details of the setup used for numerical studies are described in sufficient depth in the appendices. The three-step procedure (initialization, hyperparameter learning, inference) is clearly explained. Algorithm pseudocode aids reproducibility.

The application of GRFs to scalable GP inference is novel, though GRFs themselves were introduced recently (Reid et al. 2023, Choromanski 2023). The key theoretical results are: (1) Theorem 2 proving O(N) condition number for the covariance matrix and (2) Lemma 1 showing $O(N^{3/2})$ complexity for CG solves. The paper presents numerical studies demonstrating 50× speedups on graphs with <10K nodes and  Bayesian optimization on graphs with >1M nodes.

Overall, the paper presents solid methodological work demonstrating that GRFs enable scalable GP inference on large graphs.

**Weaknesses:**

The theoretical development is largely sound. Theorem 1 (from prior work) provides useful concentration inequalities that the paper builds upon Theorem 2's proof for $O(N)$ condition number follows from standard operator norm arguments. The constant 'c' must be finite for the concentration bounds to hold. While the authors claim this is a “mild assumption,” it effectively requires the spectrum of W to satisfy a specific decay rate. This could be made more explicit.

The analysis in Lemma 1 that claims $O(N^{3/2})$ complexity for CG is rather crude for a formal theoretical result. In practice, CG is terminated when the residual falls below a specified tolerance ($\epsilon$), requiring $O(\sqrt{\kappa} log(1/\epsilon))$ iterations. The paper does not specify what tolerance was used, nor does it analyze how the choice of tolerance affects the approximation quality of the posterior. The $O(N^{3/2})$ estimate is equivalent to assuming that CG iterations are terminated after $\sqrt{\kappa}$ steps.

The empirical scaling in Table 1 shows ~1.04 (nearly linear), which the authors attribute to a 'fixed iteration budget’. This trend implicitly suggests that the theoretical O(N^{3/2}) bound may not be achieved in practice without careful tuning of convergence criteria.

I was expecting to see approximation error analysis quantifying how using $\hat{K}$ instead of $K_\alpha$ affects the quality of the posterior. Since Theorem 1 provides concentration for individual entries, this should be feasible and would strengthen the contribution.

**Questions:**

- Section 3.1 mentions that the condition number bound requires $||\phi(i)||_1 \leq c$. Can you clarify if this bound depends on the choice of modulation function?
- Can the authors derive approximation bounds showing how the error bound on $||\hat{K} - K_\alpha||_F$ impacts the error in the posterior predictions?
- The numerical experiments show empirical scaling exponents of ~1.04 for training/inference (Table 1), which is nearly linear rather than N^{3/2}. What was the convergence tolerance used for CG? Was it an upper bound on the residual and/or maximum number of iterations?
- Was a preconditioner used for the CG solver?
- For the wind interpolation task where GRFs outperform exact kernels: (a) is this trend consistent across multiple runs? (b) Can you comment on whether this is due to overfitting of exact kernels or beneficial regularization from GRF sparsity? (c) does the error over the training/validation set show similar patterns?
- The Woodbury identity approach (Appendix B) trades sparsity for lower-dimensional inversions. Under what conditions is this preferable to the sparse CG approach? Computational cost and approximation quality comparisons would be valuable.

---

> ### Author Response · Authors · 2025-11-18
> **Rebuttal (and thanks for comments)**
>
> We sincerely thank the reviewer for their positive comments on the paper. We are pleased that they find it well written and organised, and consider our methodological contributions to be solid. We address all their concerns and questions below.
>
> 1. **Theoretical analysis**.
> - _Constant $c$ assumption_. The reviewer is correct to note that Theorem $2$ relies upon bounded $c$ across the class of graphs considered. We do not believe this to be too restrictive in practice for the following reasons. 1) Even without any sampling, the spectral radius of the weighted adjacency matrix must lie within the radius of convergence of the series in order for the exact kernel to converge; the condition here is just slightly stronger to account for the random walks. 2) In practical applications, one often uses a modulation function $f$ for which $f_i = 0 \forall i >= i_{m}$, whereupon $c$ is bounded by a constant for _any_ set of graphs. We also remark that this assumption does not cause problems in any of our experiments, including on a range of synthetic and real-world graphs. We have made this more explicit in the text. Thanks.
> - _CG termination._ The reviewer is likewise correct that we assume that CG terminates after $\sqrt{\kappa}$ steps, implicitly taking the same relative tolerance $\epsilon$ for all graphs. Here, $\epsilon$ is the target relative residual tolerance, which is set to be $10^{-2}$. We think this is reasonable since it follows convention in the literature [1] and gives robust empirical results, but we agree investigating how $\epsilon$ is related to the posterior approximation quality for different graphs is an interesting (but ambitious!) direction for future work. We finally note that similar papers in the literature sometimes claim a _constant_ number of iterations to ‘prove’ efficiency [2]; our results are more realistic.
>
> 2. **Relationship between concentration of entries and the quality of the posterior**. Thanks for the great question. On account of the kernel inverse, relating the pointwise concentration of $K$ to the quality of the posterior approximation is a difficult, open research question. Avron (2017) [3] shows how the quality of approximation of $K$ in the _spectral_ sense (see Eq. 2 therein) is related to accuracy guarantees of random feature based kernel methods on downstream tasks such as kernel ridge regression and kernel k-means clustering. The quality of the spectral approximation can in turn be related to the kernel estimator MSE [4]. We will add these pointers to the paper. However, a full theoretical analysis of how pointwise estimation relates to e.g. the KL divergence to the true posterior remains (to our knowledge) unknown, and would be a great topic for a future theoretical paper.

---

> > ### Author Response · Authors · 2025-11-18
> > **Rebuttal (2)**
> >
> > **Answers to specific questions**.
> >
> > 1. _How is the condition number bound related to the modulation function?_ The condition number bound depends upon $c$, which is related to $f$ via the definition on line 199.
> > 2. _Can the authors derive approximation bounds showing how the error bound on impacts the error in the posterior predictions?_ Please see above. This is an interesting open research question, but we think beyond the scope of the current work.
> > 3. _What was the convergence tolerance used for CG? Was it an upper bound on the residual and/or maximum number of iterations?_ The convergence tolerance was 1e-2, a standard choice for iterative GPs in the literature [1]. The maximum number of permitted iterations before termination was 1000, which we did not reach in practice in our experiments.
> > 4. _Was a preconditioner used for the CG solver?_ We did not use a preconditioner, but this would be a simple addition.
> > 5. _For the wind interpolation task where GRFs outperform exact kernels: a) is this trend consistent across multiple runs? b) Can you comment on whether this is due to overfitting of exact kernels or beneficial regularization from GRF sparsity? c) does the error over the training/validation set show similar patterns?_
> >
> > **(a) Consistency across runs:** Yes. We repeated both the wind and traffic experiments across three independent seeds (train/test splits and model initialisation). In all cases we observed the same qualitative behaviour: the fully-learnable GRF consistently outperforms the diffusion-shape GRF, and in the traffic dataset it also surpasses the exact diffusion kernel once the walk budget is large enough. The plotted curves already show the mean and variation across runs.
> >
> > **(b) Why GRFs outperform exact/diffusion-shape kernels:** We think the advantage comes from a combination of flexibility and inductive bias. 1. **Flexibility.** The GRF modulation function provides a richer kernel family than fixed-form diffusion kernels, while still involving only a small number of parameters (typically ~10), which prevents overfitting but allows adaptation to the data. 2. **Sparse, locality-biased inductive structure.** GRFs only create correlations when random-walk trajectories actually connect two nodes. This emphasises local neighbourhood structure while still allowing occasional long-range interactions. Dense diffusion kernels lack this constraint and may form spurious global correlations on noisy physical fields.
> >
> > A related observation supporting this: in the traffic dataset, marginal-likelihood
> > optimisation of the *exact* diffusion kernel consistently drives the length-scale $\beta$ to very large values, effectively flattening the kernel and causing oversmoothing. GRFs cannot collapse into such globally correlated behaviour because their random-walk sparsity prevents enforcing uniform correlations across the entire graph. This aligns with the inductive-bias explanation above.
> >
> > **(c) Training vs validation behaviour:** We primarily report test RMSE/NLPD since this is the standard metric for interpolation tasks. The optimisation of all kernels uses the marginal likelihood, which naturally discourages pathological overfitting (we did not observe instability or divergence during
> > training). Given the large training sets in both datasets, the qualitative differences we report appear mainly in test-set generalisation, which is consistent with the regularisation effect described in (b).If helpful, we can include training–validation curves in the camera-ready version.
> >
> > 6. _The Woodbury identity approach (Appendix B) trades sparsity for lower-dimensional inversions. Under what conditions is this preferable to the sparse CG approach?_ Thanks for the great question. The Woodbury approach is included as an interesting alternative to using sparse solvers, as a suggestion to prompt future research in efficient graph-based GPs. We think a full exploration of its respective strengths and weaknesses is beyond this paper’s scope. Our intuition is that the sparse version without dimensionality reduction will work better in practice – it will be interesting to test this.
> >
> > We again thank the reviewer for their time and thoughtful questions, and warmly invite them to respond.
> >
> > [1] When are Iterative Gaussian Processes Reliably Accurate? Maddox et al., OPTML 2021
> > [2] GPyTorch: Blackbox Matrix-Matrix Gaussian Process Inference with GPU Acceleration, Gardner et al., NeurIPS 2018
> > [3] Random Fourier Features for Kernel Ridge Regression: Approximation Bounds and Statistical Guarantees, Avron et al., ICML 2017
> > [4] The Geometry of Random Features, Choromanski et al., AISTATS 2018

---

### Meta-Review · Area_Chair_TMHf · 2026-01-07

**Summary:**

The reviews on this paper are generally positive -- reviewers felt that it was well-written, they thought the theoretical results are sound, and that in general the empirical evaluation is solid and demonstrates effectiveness of the method.

I took a look at the paper and personally, I find the contribution incremental. Random features have been used extensively in approximate GP inference outside the graph domain, and the idea here is basiaclly identical to prior work. It is more an application of known techniques than a novel research contribution. While several reviewers praise the theoretical results of the paper, Theorem 1 is from prior work and Theorem 2 is immediate -- it is just observing that for any n x n matrix with entries bounded by O(1), its spectral norm is bounded by O(n). Lemma 1 is also immediate from results on the convergence of CG. There is no deep theory provided here.

Overall, given my own assessment as compared to the reviewers, I find this paper borderline. But am recommending acceptance inline with the reviewers.

**Reviewer Concerns:**

One concern brought up was that the paper does not compare to other popular kernel approximation approaches, like inducing point methods. This was addressed clearly by the authors, who point out that in the graph setting, it is not clear how to apply such methods efficiently, as we cannot efficiently compute individual entries of the kernel matrix.

Another issue pointed out by several reviewers was that the paper does not analyze how the approximation of the underlying kernel matrix translates to approximation of the posterior inference problem. I agree with the authors that this is an interesting direction but potentially a bit outside the scope of this work -- it is a challenging theoretical question that isn't even very well understood for e.g. traditional GPs.

**Reviewer Scores:**

The reviewers were fairly positive to begin with. I don't think any would have changed their scores. Perhaps RsGB or sDRg would have bumped up their scores by 1 given the authors' response about why they can't directly compare to inducing point methods.

---

### Decision · Program_Chairs · 2026-01-26

Accept (Poster)